# Human hippocampal replay during rest prioritizes weakly learned information and predicts memory performance

Anna C. Schapiro [1], Elizabeth A. McDevitt [2], Timothy T. Rogers [3], Sara C. Mednick [4] &
Kenneth A. Norman [2]

The hippocampus replays experiences during quiet rest periods, and this replay benefits subsequent memory. A critical open question is how memories are prioritized for this replay. We used functional magnetic resonance imaging (fMRI) pattern analysis to track item-level replay in the hippocampus during an awake rest period after participants studied 15 objects and completed a memory test. Objects that were remembered less well were replayed more during the subsequent rest period, suggesting a prioritization process in which weaker memories—memories most vulnerable to forgetting—are selected for replay. In a second session 12 hours later, more replay of an object during a rest period predicted better subsequent memory for that object. Replay predicted memory improvement across sessions only for participants who slept during that interval. Our results provide evidence that replay in the human hippocampus prioritizes weakly learned information, predicts subsequent memory performance, and relates to memory improvement across a delay with sleep.

[1] Department of Psychiatry, Beth Israel Deaconess Medical Center and Harvard Medical School, Boston, MA 02215, USA. [2] Princeton Neuroscience Institute and Department of Psychology, Princeton University, Princeton, NJ 08544, USA. [3] Department of Psychology, University of Wisconsin-Madison, Madison, WI 53706, USA. [4] Department of Cognitive Sciences, University of California-Irvine, Irvine, CA 92617, USA. Correspondence and requests for materials should be addressed to A.C.S. (email: aschapir@bidmc.harvard.edu)

The brain is highly active even when an organism is dis-engaged from its sensory environment[1]. There is accu-mulating evidence from the rodent literature that the hippocampus replays recent experiences during these rest peri-ods, typically measured as place cells firing in a sequence corre-sponding to an experienced trajectory of locations[2,3]. These replay events appear to be functional: they relate to later memory performance and their disruption impairs memory[4–6]. Many human studies, mainly using fMRI, likewise suggest that content from a recent experience reactivates during subsequent rest per-iods, and that this activity relates to later memory[7–17]. These studies have found replay of individual items outside the hippo-campus[7–9] and category-level reinstatement within the hippo-campus[10,14], though replay of individual items within the human hippocampus has not yet been observed.

What process determines which memories get replayed? The brain likely cannot replay every experienced event during rest, nor would it be worthwhile to do so—not all memories need or deserve further processing. There is evidence that memories are more likely to be replayed during subsequent awake rest when associated with reward or fear[14,15,17]. There is also evidence that certain kinds of memories benefit more from a period of sleep, including memories that are relevant to future behavior[18], that are emotionally laden[19], and that are weaker in strength[20–27], c.f. [28,29]. We observed this prioritization of weaker memories in prior work using the present paradigm, where objects exposed least frequently during training benefitted the most from a nap[30]. These sleep studies do not directly demonstrate that replay prioritizes weaker (or other types of) memories, as replay was not measured, but the ubiquity of sleep replay and its association with memory improvement[31,32] suggest that prioritized replay is a potential mediator of the behavioral benefits. While there have been many studies assessing the relationship of rest replay to subsequent memory, there has not yet been a direct test of how initial memory strength relates to subsequent replay.

The current study assesses how replay is prioritized on the dimension of memory strength, and tests the effects of such replay on subsequent memory, using a property-inference task developed in prior work[30]. Participants learned the features of 15 "satellite" objects belonging to three categories (Fig. 1), where satellites in the same category shared most of their features. Those assigned to a Sleep group participated in a first session in the evening and a second session the next morning, while those assigned to a Wake group completed their first session in the morning and second session that evening. In Session 1, we (1) taught participants the features of the satellites, (2) tested their memory for these features, (3) measured the neural response generated by each of the satellites in the fMRI scanner, and then (4) used pattern analysis to assess whether individual items were replayed by the hippocampus during a rest period in the scanner. In the second session, we again measured neural responses to the individual satellites and assessed replay during rest. After the rest period, we tested memory for the satellites a second time. Note that in Session 1, the memory test preceded the rest period whereas in Session 2, the rest period preceded the memory test.

This design allowed us to answer four questions critical to understanding the role of hippocampus in the consolidation of object knowledge:

1. Are representations of recently learned individual items replayed in the human hippocampus during quiet rest? Prior literature in humans and rodents suggests that this occurs, but it has not yet been observed at the resolution of individual items in humans.
2. In Session 1, where the memory test precedes the rest period, does probability of replay during rest relate to the strength of

a memory? It is possible that weaker memories are prioritized for replay, as suggested by the sleep literature; alternatively, stronger memories may be more likely to persist into subsequent rest.
3. In Session 2, where the rest period precedes the memory test, does replay of individual items in the hippocampus predict subsequent memory? Prior literature in humans and rodents suggests a positive relationship, though this has not yet been assessed for individual hippocampal memories in humans.
4. Does the relationship between wake replay and memory improvement across the delay between sessions relate to the presence of intervening sleep? Replay measured during awake rest periods may reflect (or perhaps influence[33,34]) the processing that continues to occur in the intervening period between sessions, and this processing may be especially beneficial over sleep[35].

We find evidence for replay of individual items in the human hippocampus during quiet rest, with a prioritization of objects that had been remembered less well. We also find that replay of an object predicts better subsequent memory for that object, and that replay predicts memory improvement across a delay only if the delay includes sleep. The findings suggest that replay adap-tively focuses on memories most in need of help and that memories benefit from this replay.

## Results

**Sleepiness survey**. Karolinska Sleepiness Scale[36] (KSS) sleepiness scores did not differ across Sleep and Wake conditions for the beginning of Session 1 (mean Sleep = 4.042; mean Wake = 4.417; $t[22] = 0.462$, $p = 0.649$), end of Session 1 (mean Sleep = 6.167, mean Wake = 5.333, $t[22] = 1.299$, $p = 0.207$), beginning of Session 2 (mean Sleep = 4.083, mean Wake = 4.083, $t[22] = 0$, $p = 1$), or end of Session 2 (mean Sleep = 4.250; mean Wake = 5.208, $t[22] = 1.101$, $p = 0.283$), suggesting that there were no alertness differences between groups due to time of day.

**Training and one-back performance**. In learning about the satellites outside of the scanner, participants trained for an average of 121.5 trials (SD = 83.2), including repetition trials for incorrect choices. Average proportion correct on the last training block was 0.747 (SD = 0.088). Detection performance on a one-back cover task during measurement of item representations in the scanner was excellent (mean A′ = 0.916, SD = 0.060).

**Test performance**. Each satellite has shared features: the class name and the parts shared among members of its category, and unique features: the code name and the part unique to that satellite (except for the category prototype, which has no unique parts). Performance was assessed separately for unique and shared features, and for features belonging to a satellite that was not encountered during training (a novel satellite). Performance on the first test, prior to the sleep or wake intervention, was not different for subjects in the Sleep vs. Wake groups for unique, shared, or novel item features ($p$'s > 0.564), suggesting that time of day does not influence performance on this task (as in our pre-vious work with this paradigm[30]). Performance was also not different within each group for unique and shared features (mean Sleep unique = 0.701, mean Sleep shared = 0.677, $t[11] = 0.444$, $p = 0.666$; mean Wake unique = 0.695, mean Wake shared = 0.715, $t[11] = 0.686$, $p = 0.507$). The tests were designed to minimize learning within the test phase. To verify that no within-test learning occurred, we ran an ANOVA on the data from both sessions, with feature type, session, and first vs. second half of the test phase as factors. There were no main effects or interactions

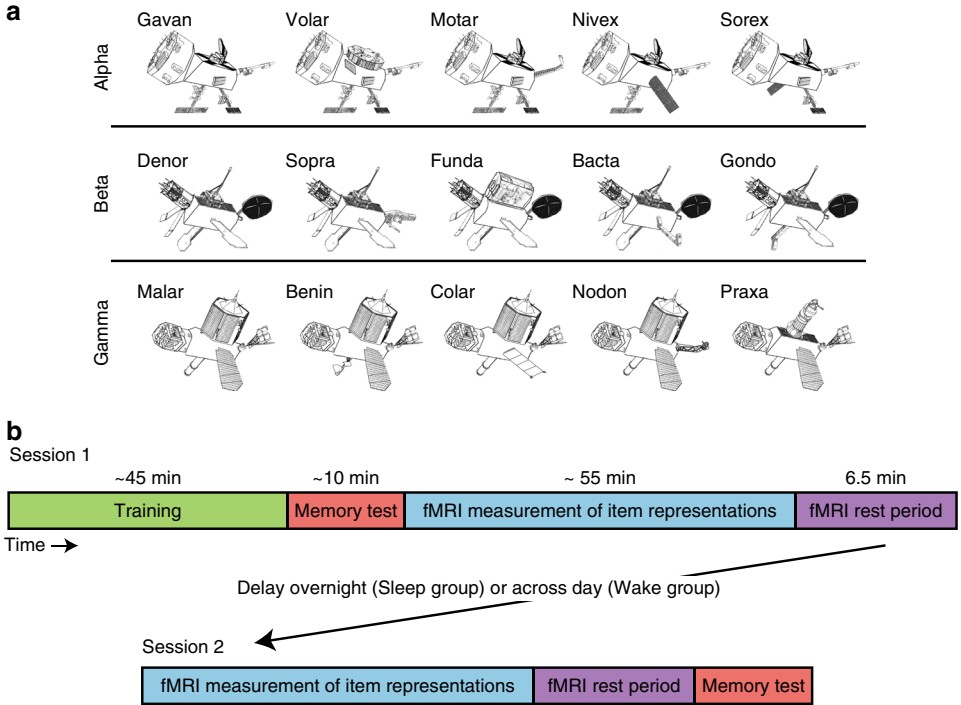

**Fig. 1** Stimuli and design. **a** Examples of stimuli presented from the three classes Alpha, Beta, and Gamma, labeled with unique code names. Satellites were built randomly for each participant, using the same category structure. **b** Sessions 1 and 2 procedures for all participants, with delay between sessions either overnight or across day depending on group

for first vs. second half of the test phase ($p$'s > 0.101), indicating no evidence of learning within the test phases.

Based on our prior work using this paradigm[30], we predicted that the Sleep group would improve in memory for shared features from the first to second session, the Wake group would decline in memory for shared and unique features, and the Sleep group would improve more than the Wake group for shared and unique features. We thus employed one-tailed $t$-tests in these analyses. As expected, we found a reliable difference between Sleep and Wake groups in improvement from the first to second session in overall memory (Fig. 2; $t[22] = 2.110$, $p = 0.023$, one-tailed, Cohen's $d = 0.862$), and a reliable difference between groups for unique visual ($t[22] = 2.190$, $p = 0.020$, one-tailed, $d = 0.894$) and shared visual features ($t[22] = 1.995$, $p = 0.029$, one-tailed, $d = 0.814$).

The Sleep group showed significant improvement in overall memory (collapsing across shared and unique features) from Session 1 to Session 2 tests (Fig. 2; mean improvement in proportion correct = 0.057; $t[11] = 2.498$, $p = 0.015$, one-tailed, $d = 0.721$), while the Wake group showed a nonsignificant decrement in performance (mean = −0.022, $t[11] = 0.731$, $p = 0.240$, one-tailed, $d = 0.211$). As in our prior study, memory for shared features in the Sleep group increased reliably, while memory for unique features stayed constant (mean change for shared = 0.137, $t[11] = 4.187$, $p = 0.001$, one-tailed, $d = 1.209$; mean change for unique = −0.022, $t[11] = 0.693$, $p = 0.503$, $d = 0.200$). In the Wake group, there was a nonsignificant improvement in memory for shared features (mean = 0.049, $t[11] = 1.333$, $p = 0.896$, one-tailed, $d = 0.385$), and a decrease in memory for unique features (mean = −0.092, $t[11] = −2.081$, $p = 0.031$, one-tailed, $d = 0.601$). Note that participants were exposed to the visual features of the satellites in the scanner between the two behavioral tests, which could have affected absolute levels of change from Session 1 to Session 2 within each

group, so we have plotted visual and verbal (name) features separately in Fig. 2. However, this issue does not affect Sleep/ Wake comparisons, as visual feature exposure was matched across these conditions.

Change in novel item feature performance was similar for the two groups, as we had found in our prior study. There was a marginal increase in performance in both groups (Sleep mean = 0.120, $t[11] = 1.995$, $p = 0.071$, $d = 0.576$; Wake mean = 0.074, $t[11] = 2.000$, $p = 0.071$, $d = 0.577$). Since we did not measure the representation of novel items in the scanner, we do not consider these items in further analyses.

**Session 1 correlation between memory and subsequent replay.** In Session 1, the memory test came before the rest period. To investigate the relationship between Session 1 performance and subsequent hippocampal replay, we first assessed reactivation of individual satellite representations. We created multivoxel templates for each satellite based on activity in the hippocampus during the item measurement period (Fig. 1b), where satellites were presented 32 times each in a pseudorandom order. We then correlated these templates with rest period activity (Fig. 3a), treating strong correlations as (potential) replay events (Fig. 3b). We then correlated, for each subject, memory for each satellite in Session 1 with the sum of hippocampal replay activity for that satellite during Session 1 (Fig. 3c). We assessed the mean value of these correlations across subjects. We collapsed across Sleep and Wake groups for these analyses. Collapsing across left and right hippocampus, we found reliably negative correlations (mean = −0.145, $t[23] = 3.442$, $p = 0.002$, $d = 0.703$), which were significant in both left (mean = −0.113, $t[23] = 2.255$, $p = 0.034$, $d = 0.460$) and right (mean = −0.177, $t[23] = 2.664$, $p = 0.014$, $d = 0.544$) hippocampus individually, indicating that replay was strongest for the satellites that were remembered worst on the preceding test (Fig. 4a).

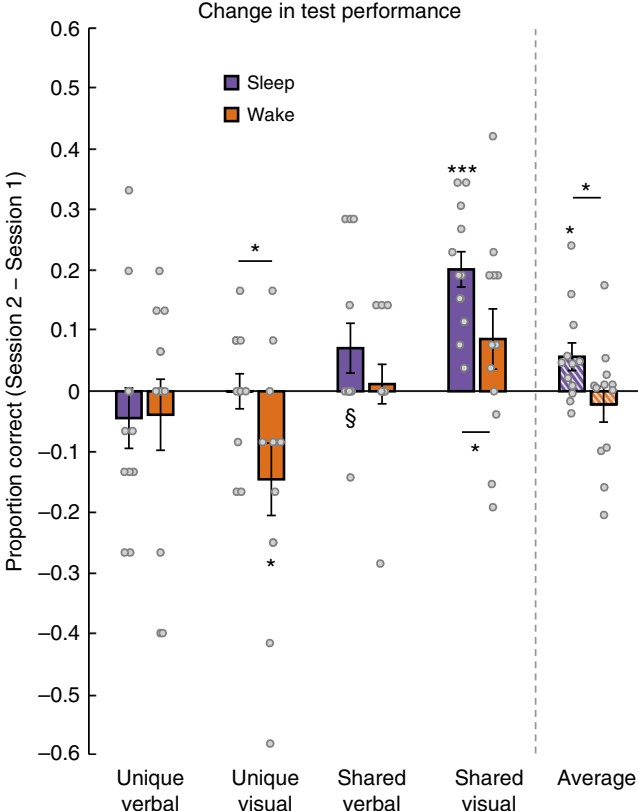

**Fig. 2** Behavioral results. Change in proportion correct from the first to second session for Sleep and Wake group, for different feature types and the average across feature types; $^§p < 0.1$, $^*p < 0.05$, $^{***}p < 0.001$, one-tailed $t$-test. Asterisks above horizontal lines show significant differences between conditions; asterisks without bars indicate where conditions differ from zero. Gray circles indicate performance of individual subjects. Error bars denote ±1 SEM; $n = 12$ in each group

cannot be computed, in cases where subjects performed at floor or ceiling on all items within that item type (though there are still a few cases where subjects performed perfectly and had to be excluded from a particular analysis, as reflected in the variation in $df$ below). We found no effects of any of these variables (Fig. 4b–d). In both sessions, left was not different than right hippocampus (Session 1: $t[23] = 0.773$, $p = 0.447$; Session 2: $t[23] = 1.453$, $p = 0.160$), the Sleep group was not different than Wake (Session 1: $t[22] = 0.368$, $p = 0.717$; Session 2: $t[22] = 0.764$, $p = 0.453$), verbal was not different than visual (Session 1: $t[22] = 1.001$, $p = 0.328$; Session 2: $t[21] = 0.756$, $p = 0.458$), and shared was not different than unique (Session 1: $t[23] = 0.203$, $p = 0.841$; Session 2: $t[20] = 0.693$, $p = 0.496$). Note that the lack of difference between Sleep and Wake groups in Session 1 suggests that time of day did not influence replay.

We also ran mixed effects models for each session with Sleep vs. Wake as an across-subjects factor and verbal vs. visual, left vs. right hippocampus, and unique vs. shared as within-subjects factors. Mixed effects models can gracefully handle missing data, allowing us to assess potential interactions between factors. We did not find evidence for reliable interactions between any factors in Session 1. We did find evidence for an interaction between unique vs. shared and verbal vs. visual information in Session 2, with replay more positively associated with visual shared features than verbal shared features, and more positively associated with verbal unique features than visual unique features ($X^2$ for models with vs. without interaction = 5.70, $p = 0.017$). These analyses should be interpreted with caution, as power for assessing interactions in this study is low[37]. Potential interactions between these variables will need to be further assessed in future work.

**Session 2 correlation between replay and subsequent memory.** In Session 2, the rest period came before the memory test. We did the same analysis as above, now using the Session 2 item measurement period to create multivoxel templates to identify replay events during the rest period, and correlating Session 2 replay with subsequent Session 2 memory. Here we found positive correlations that were reliable when collapsing across the left and right hippocampi (mean = 0.112; $t[23] = 3.262$, $p = 0.003$, $d = 0.666$) and also in the right hippocampus on its own (mean = 0.165, $t[23] = 3.841$, $p = 0.0008$, $d = 0.784$; mean left hippocampus = 0.058, $t[23] = 1.021$, $p = 0.318$, $d = 0.208$). This indicates that satellites that were replayed more in Session 2 were subsequently better remembered (Fig. 4a). The correlations were more positive in Session 2 than Session 1 (collapsing across left and right hippocampi: $t[23] = 4.785$, $p = 0.00008$, $d = 0.977$; left: $t[23] = 2.039$, $p = 0.053$, $d = 0.416$; right: $t[23] = 5.164$, $p = 0.00003$, $d = 1.054$).

**Effects of group, hemisphere, and feature type.** To assess whether these effects differed by Sleep vs. Wake group, left vs. right hippocampus, verbal (class and code names) vs. visual information, and unique vs. shared features, we tested within each session whether the memory–replay correlations were different for each variable collapsing across the other variables. We opted for this approach because breaking the results down by combinations of these features leads to cells with correlations that

**Specificity of the replay to individual satellites.** We ran the same replay analyses after scrambling the labels of the satellites during estimation of the template representations. The scrambling procedure should render the templates meaningless and result in less replay detection and no relationship with behavior. Indeed, after scrambling, there was significantly less detected replay overall (Session 1: $t[23] = 2.520$, $p = 0.019$; Session 2: $t[23] = 2.790$, $p = 0.010$, collapsed across right and left hippocampus) and no relationship with behavior (Session 1: mean = −0.020, $t[23] = 0.404$, $p = 0.690$; Session 2: mean = 0.0005, $t[23] = 0.011$, $p = 0.991$). This suggests that the measured replay activity in the analyses above reflects the reinstated representation of specific satellites. To further assess whether we were observing replay of individual satellites as opposed to a more abstract representation of a satellite's category, we ran two additional control analyses. One analysis shuffled the labels of satellites from the same category during the calculation of template–rest period correlations, and again resulted in no evidence of meaningful replay–behavior relationships (Session 1: mean = 0.009, $t[23] = 0.440$, $p = 0.664$; Session 2: mean = −0.034, $t[23] = 1.674$, $p = 0.108$). The Session 1 correlations were significantly higher than in the correctly aligned data ($t[23] = 2.926$, $p = 0.008$) and the Session 2 correlations were significantly lower ($t[23] = 4.179$, $p = 0.0004$). These results provide strong evidence that the templates are matching up with the neural representations of specific items within a category. Finally, as a positive control, we assessed whether the results would remain the same if replay thresholds were calculated with respect to other items from the same category (i.e., a satellite representation must be reinstated more than other members of its category to count as being replayed), as opposed to all possible items. The results were indeed qualitatively similar. The mean correlation between replay and memory in Session 1 in left hippocampus was −0.124 ($t[23] = 2.003$, $p = 0.057$) and in right hippocampus was −0.086 ($t[23] = 1.399$, $p = 0.175$). In

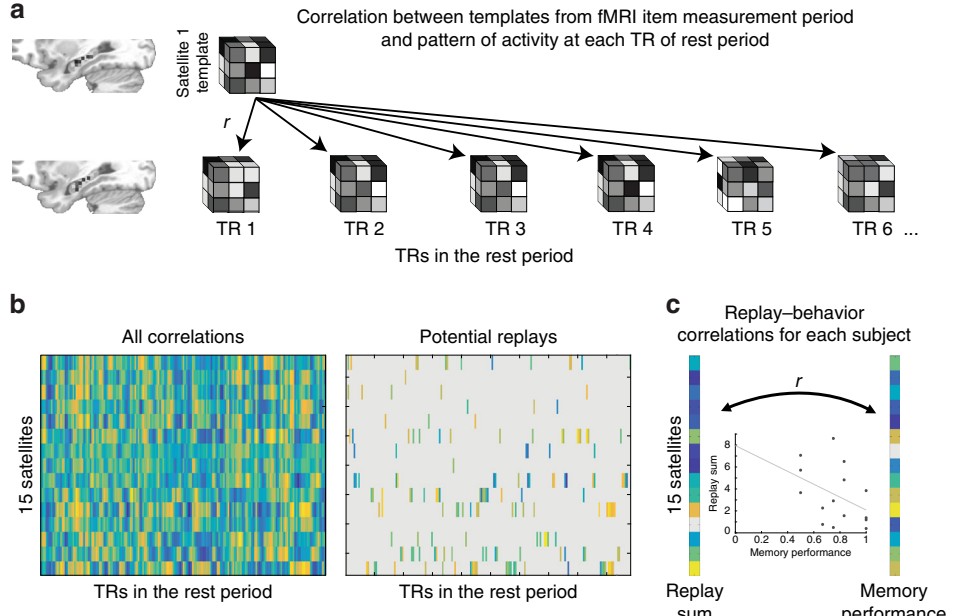

**Fig. 3** Replay analysis. **a** Each satellite's template consists of a pattern of beta weights across all voxels in the hippocampus. The template is correlated with the preprocessed pattern of activity across all voxels in the hippocampus at each TR of the rest period. **b** This results in a matrix of correlation values between all templates and all rest period TRs, which is then thresholded to reflect "potential replays". **c** The potential replay activity for a given template is summed across TRs and then correlated with memory performance. The inset scatterplot shows data for one representative subject's Session 1 correlation

Session 2, the mean correlation between replay and memory in left hippocampus was $-0.027$ ($t[23] = 0.332$, $p = 0.743$) and in right hippocampus was $0.184$ ($t[23] = 2.814$, $p = 0.010$). The difference between the correlations in Sessions 1 and 2 was reliable in right hippocampus ($t[23] = 3.398$, $p = 0.003$; left hippocampus: $t[23] = 0.874$, $p = 0.391$). Overall, these analyses suggest that we are observing replay of individual satellite representations.

**Cross-session replay–behavior relationships**. Session 1 replay was correlated with Session 2 behavior (mean $= -0.136$, $p = 0.039$, collapsed across right and left hippocampus; correlation between Session 2 replay and Session 1 behavior mean $= 0.077$, $p = 0.135$), which is likely due to the fact that behavior is correlated across sessions (mean correlation $= 0.394$, $p < 0.0001$). This relationship did not withstand regressing out within-session behavior (mean coefficient $= -0.027$, $p = 0.175$; for Session 2 replay, mean coefficient $= -0.651$, $p = 0.964$), whereas within-session replay–behavior relationships largely remained when regressing out other-session behavior (mean coefficient for Session 1 behavior predicting Session 1 replay after regressing out Session 2 behavior $= -1.22$, $p = 0.066$; mean coefficient for Session 2 behavior predicting Session 2 replay after regressing out Session 1 behavior: $1.339$, $p = 0.051$). Memory and replay are thus most closely related within the same session.

**Cross-session replay–replay relationships**. For each individual, we calculated the correlation between amount of replay of each of the 15 satellites in Session 1 and amount of replay of each of the 15 satellites in Session 2. These correlations were not reliably different from zero across subjects ($t[23] = 0.871$, $p = 0.393$, collapsed across right and left hippocampus).

**Correlation between replay and memory change across sessions**. We next separated Sleep and Wake groups and asked whether replay in either session relates differently to behavioral

change across sessions for subjects who slept vs. did not sleep between sessions. Because prior work strongly predicts that replay should improve performance over sleep, we employed one-tailed $t$-tests in the analysis of the Sleep group, and in the comparison between Sleep and Wake groups. We correlated replay for each satellite in each session with change in performance for that satellite (Session 2—Session 1 performance). We found that, across hemispheres of the hippocampus and across sessions, replay in the Sleep group was indeed positively related to memory improvement (Fig. 5; mean $= 0.093$, $t[11] = 2.146$, $p = 0.028$, one-tailed, $d = 0.620$), and that this effect was reliably greater than in the Wake group ($t[22] = 2.522$, $p = 0.010$, one-tailed, $d = 1.030$), which had a numerically negative relationship (mean $= -0.066$, $t[11] = 1.441$, $p = 0.177$, $d = 0.416$). For particular sessions and hemispheres, there was a marginal negative effect in left hippocampus in Session 1 in the Wake group (mean $= -0.134$, $t[11] = 1.814$, $p = 0.097$, $d = 0.524$), which was significantly lower than in the Sleep group ($t[22] = 1.782$, $p = 0.044$, one-tailed, $d = 0.728$), and a positive effect in the Sleep group in the right hippocampus in Session 2 (mean $= 0.186$, $t[11] = 2.289$, $p = 0.022$, one-tailed, $d = 0.661$).

We also tested the idea that replay in Session 1 in the Sleep group might be more directly predictive of Session 2 behavior (after accounting for Session 1 behavior) than replay in the Wake group. This analysis assesses cross-session replay–behavior relationships, as described above, separately for the Sleep and Wake groups. We found evidence in support of this idea: Session 1 replay predicts Session 2 behavior more strongly in the Sleep group than the Wake group ($t[22] = 1.847$, $p = 0.039$, one-tailed, $d = 0.754$).

**Representation of category structure**. To test whether the hippocampus was sensitive to the category structure of the stimuli, we assessed whether satellites from the same category were represented more similarly than satellites from different categories, collapsing across groups and sessions. The hippocampus

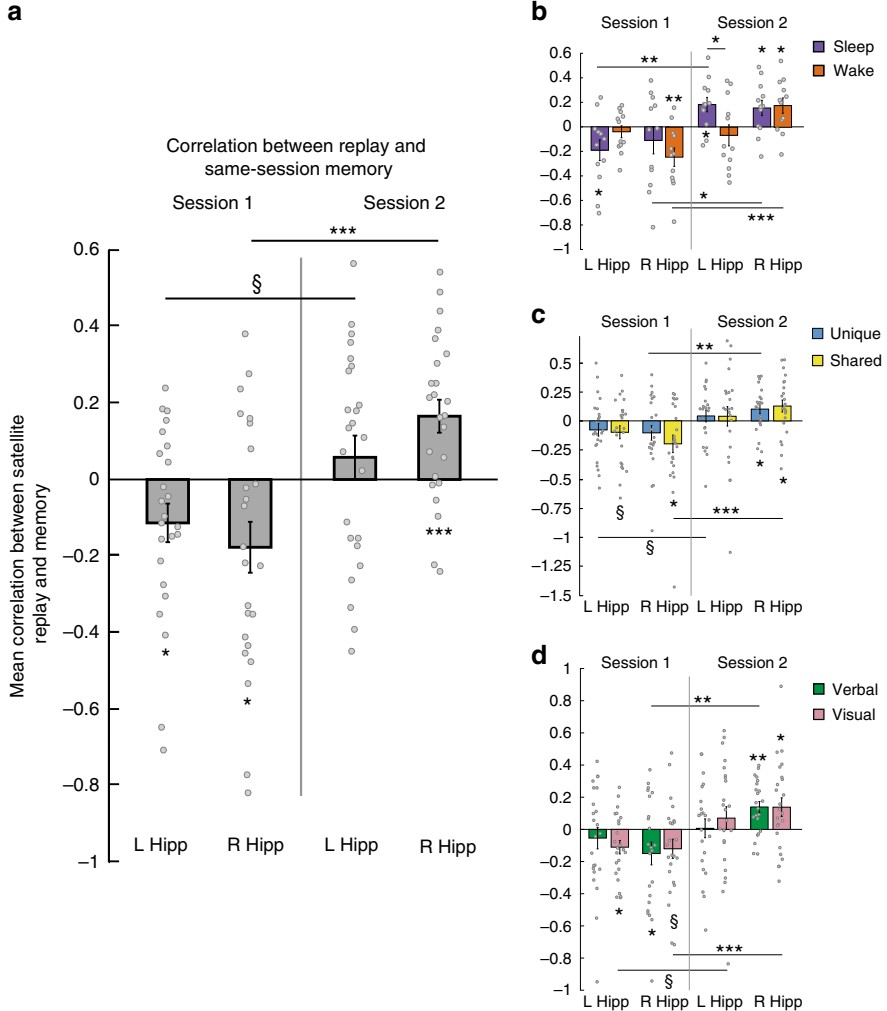

**Fig. 4** Within-session replay–behavior relationships. **a** Correlation between memory in Session 1 and replay in Session 1, and between memory in Session 2 and replay in Session 2, in left (L Hipp) and right (R Hipp) hippocampus. **b** Same data, broken down by Sleep and Wake groups. **c** Same data, broken down by memory for unique and shared features. **d** Same data, broken down by memory for verbal and visual features. $^{\S}p < 0.1$, $^*p < 0.05$, $^{**}p < 0.01$, $^{***}p < 0.001$, *t*-test. Error bars denote ±1 SEM; $n = 24$ for all bars except **b**, where $n = 12$ for each group

overall did not show any sensitivity to category structure (both hippocampi: $t[23] = 0.341$, $p = 0.737$; left: $t[23] = 0.845$, $p = 0.407$; right: $t[23] = 1.600$, $p = 0.123$). Considering separate subfields, however, a reliable effect was observed in CA1 (both hemispheres: $t[23] = 2.336$, $p = 0.029$, $d = 0.477$; left: $t[23] = 1.685$, $p = 0.106$, $d = 0.344$; right: $t[23] = 2.345$, $p = 0.028$, $d = 0.479$) but not CA2/3/DG (both hemispheres: $t[23] = 0.612$, $p = 0.547$, $d = 0.125$; left: $t[23] = 1.068$, $p = 0.297$, $d = 0.218$; right: $t[23] = 0.359$, $p = 0.723$, $d = 0.073$). Overall similarity between stimuli was also higher in CA1 relative to CA2/3/DG ($t[23] = 2.349$, $p = 0.028$, $d = 0.480$). These findings are consistent with a previous proposal about the role of CA1 in representing structured information[38]. None of the results differed by session or group ($p$'s > 0.196). These subfield results should be treated as exploratory, as our functional scanning resolution was too low to permit confident assessment of subfield activity.

**Whole-brain analyses**. We ran exploratory whole-brain searchlights looking for other regions where item replay during Session 1 or Session 2 might relate to behavior within that session, or regions where replay during Session 1 or Session 2 might relate to

the change in performance from Session 1 to 2. No regions survived multiple comparisons correction.

We also ran a searchlight to test whether areas outside the hippocampus might represent the category structure (as found in hippocampal subfield CA1). We found two significant clusters, both associated with visual processing, identified according to a probabilistic atlas of the visual system:[39] One in visual cortex (corrected $p = 0.0002$), spanning V1–V4, and another in frontal cortex (corrected $p = 0.031$), partially overlapping with the frontal eye fields and falling within regions of the inferior frontal sulcus involved in spatial vision[40]. Responses in these areas likely reflect the fact that objects in the same category share visual features. There were no differences in category structure between sessions or groups in this whole-brain analysis.

We next tested whether there were any changes in overall, univariate activity level from the first to second session. While there were no effects that survived cluster correction for the Wake group, or for Sleep contrasted with Wake, we did find a reliable cluster in the medial prefrontal cortex (mPFC) that increased in activity from Session 1 to 2 in the Sleep group (corrected $p = 0.030$; coordinates of center of gravity: 45.2, 87.8, 35.8; extent = 521 voxels).

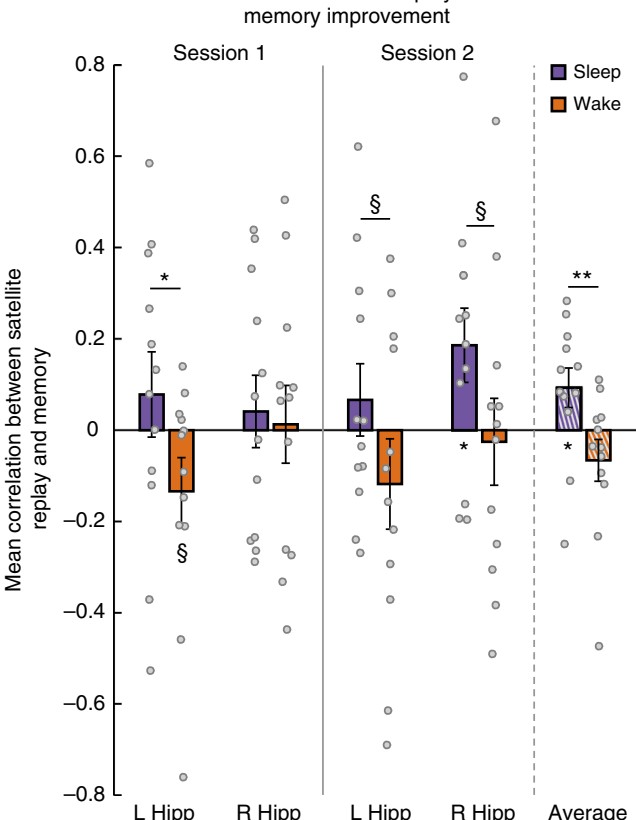

**Fig. 5** Across-session replay–behavior relationships. Correlation between replay in each session and improvement in memory from the first to second session, as well as the average for each group across sessions; $^{\S}p < 0.1$, $^{*}p < 0.05$, $^{**}p < 0.01$, $t$-test, one-tailed for comparing Sleep to zero and Sleep to Wake. Error bars denote ±1 SEM; $n = 12$ in each group

**Assessing the role of attention**. The experiment was designed such that the item measurement and rest periods occurred in the same order in both sessions, in case the order of these two experiment elements might have consequences for the detection of replay events. This design decision required the item measurement period to come between the memory test and rest period in Session 1, which introduces the possibility that unassessed new learning could occur during the measurements. This leads to a potential alternative explanation for the negative correlation between memory and subsequent replay in Session 1: If participants attend more during the measurement period to the stimuli that they had just performed poorly on, this could cause persistence of the representations of those attended satellites into the subsequent rest period. To test this idea, we assessed whether satellites with worse average memory performance in Session 1 had higher BOLD activity in a canonical attention network ("attention" network from neurosynth.org using reverse inference: *attention_pFgA_z_FDR_0.01.nii*) during the subsequent measurement period. We found no evidence for a relationship (mean correlation = 0.014, $t[23] = 0.253$, $p = 0.803$). We also tested for this effect in the hippocampus and found no relationship there (mean correlation = −0.003; $t[23] = 0.060$, $p = 0.952$). Next, we ran the analysis at every voxel in the brain and assessed whether any areas showed reliable cluster-corrected results. There were no reliable clusters looking across the whole brain, nor with small volume correction within the attention network (all $p$'s >0.686). We therefore conclude that attention to poorly remembered stimuli

during the measurement period is unlikely to be driving the negative relationship between memory and replay in Session 1.

## Discussion

How does hippocampal replay during quiet rest relate to previous learning and to subsequent memory? To address these questions, we taught participants the features of 15 objects and assessed replay of individual object representations before and after memory tests. We did extensive repeated exposure of the objects during fMRI scanning to maximize our ability to create high-fidelity templates that could be used to track replay. We found in the first session, where a memory test preceded a rest period, that objects that were initially remembered poorly were subsequently replayed more in the hippocampus, while in the second session, where the rest period preceded the memory test, objects that were replayed more were subsequently better remembered. Furthermore, replay was related to change in memory from the first to second test only for participants that slept between the two sessions. These results address the four questions raised in the introduction, with important implications for our understanding of consolidation of object knowledge.

While prior work has shown item-specific replay outside the hippocampus[7–9] and coarser replay within the hippocampus[10,14,41], the current results provide evidence for item-specific replay in the human hippocampus. Although the word "replay" is sometimes reserved for the sequential replay of place cell activity observed in rodents, we have used the term more broadly to mean neural reactivation of recent experience, including that observed in fMRI. In using the same term as in the rodent literature, we posit that the reactivation we observe has fundamental properties in common with that observed in the rodent literature. Future work will determine whether these phenomena are indeed related enough to continue to deserve the same nomenclature.

Our demonstration of item-specific replay is grounded in the observed correlations (computed across-items, within-subjects) between replay and memory performance and bolstered by control analyses demonstrating that the replay is not generalized across items within the same category. This replay seems likely to reflect incidental, endogenous activity, as participants did not report trying to rehearse the satellites and were not told that there would be a second memory test. These reports were collected at the end of the experiment, so it is possible that participants did do some strategic rehearsal that they later forgot. Future work could assess the degree of implicitness of the replay more directly by asking participants for reports throughout the experiment. Another question for future work is whether the mere possibility of future memory tests was a driver of either implicit or explicit replay, which could be assessed in a design that leads participants to believe that their memory will not be tested again.

Prior fMRI studies showing a positive relationship between replay and subsequent memory[7–16] fit with the idea that replay causes better memory. However, these results can also be explained in terms of stronger memories being replayed more, without there necessarily being a causal relationship between replay and subsequent memory. Our results from the second session fit with this latter story, but the results from the first session do not. Instead, the negative correlation between replay and preceding memory in the first session suggests the use of a prioritization process that focuses on weaker memories. This is consistent with prior findings from this paradigm[30], where weaker items benefitted from a short period of sleep (whereas overnight sleep benefitted all items). Taken together, these findings suggest that the brain's first priority is to process weaker information. One study in rodents found that more replay

occurred for less frequently experienced information[42], perhaps reflecting a similar process. Other studies in humans have found prioritization of weaker memories in working memory replay[43] and more benefit of awake cued reactivation for weaker memories[44]. Prioritization of weaker information is also consistent with the finding that encoding difficulty predicts the occurrence of more sleep spindles (which are associated with replay) during a subsequent nap[45] and the general tendency for sleep to benefit information that is more weakly encoded[20–27], c.f. [28,29]. It is important to note that memory strength is not the only dimension that can drive prioritized wake replay; other known dimensions include fear[17] and reward[14,15,46].

One human fMRI study found that activity patterns in the hippocampus that were strongest during encoding (explaining the most variance in a Principal Component Analysis) persisted into rest and correlated with subsequent memory[10]. This may seem to contrast with our study and others, but these strong patterns did not necessarily correspond to strongly encoded items—they could in principle reflect a tagging process occurring for weaker memories.

Because the memory test always came between the training and rest periods in Session 1, we do not know whether the prioritization process identified weak memories during the training phase or the test phase. Future studies could probe this by having a rest period immediately after the training phase. Participants did not receive feedback during the test phase in our study, so if weak memories were identified during that phase, some process sensitive to the strength of the representation generated during test must have provided the signal that subsequently led to increased wake replay. Notably, there was no learning within the test phases (e.g., for the Session 1 test, mean accuracy numerically decreased from the first half to the second half, dropping from 0.72 to 0.70), so this signal does not have an immediate memory benefit for poorly learned items, but instead requires later offline processing.

It seems unlikely that the relationship between memory strength and replay likelihood is strictly monotonic. If an experience is barely encoded at all, there may not be a strong enough representation of that experience to support any replay. With respect to the full range of encoding strength possible in everyday life, ours and other studies with many learning trials are likely sampling from the medium to high range of memory strength. It would be interesting for future work to map out a wider range of memory strength to assess whether replay likelihood is indeed an inverted U-shaped function of strength.

In Session 2, items that were replayed more showed better subsequent memory. This result is consistent with accumulating findings from both the rodent[2,5] and human literature[7–17,47] suggesting a functional role for hippocampal replay (i.e., improved consolidation), although—for reasons discussed above —this correlation unto itself is not evidence that replay causes better subsequent memory.

Wake replay in both sessions was more strongly associated with memory improvement for participants who slept than for those who remained awake between sessions. Thus, while wake replay may have its own benefits, it may also lay the groundwork for subsequent sleep-dependent consolidation processes. One hypothesis suggested in other work[33,34] is that processing during wake rest serves to "tag" memory traces for subsequent replay during sleep. Alternatively, it may be that some other factor occurring prior to rest (either during learning, or during the memory test) drives prioritization during both wake and sleep replay, and that the larger effects observed in the Sleep group arise because those participants encountered less wake-based interference, had more replay overall (during sleep), or that the replay that occurred during sleep was especially beneficial. In either case, some process must occur prior to the rest period that constrains which representations are revisited later.

Consistent with our prior study[30], memory for the satellite objects, and in particular the features shared across members of a category, was better after a night of sleep. These results may reflect an increased reliance on cortical areas as a result of systems consolidation[48], as cortical areas employ more overlapping representations than the hippocampus and should therefore facilitate memory for shared features[30,49]. One cortical region often implicated as a consolidation site is the mPFC[50], and our whole-brain univariate analysis found increased activation across sessions in that region for the Sleep but not the Wake group. We did not find evidence for multivariate changes—in replay or representational similarity—in cortical regions, which may mean multivariate changes did not occur on this timescale, or it may simply be a null result due to methodological or power limitations. The univariate finding suggests that the mPFC is overall more engaged by the satellites after a period of sleep, but leaves open whether there are changes in its representational content. We did not find any evidence that the hippocampus was less involved in this task from the first to second session in the Sleep group, c.f. [51], though any decreased reliance on the hippocampus may take more time.

We did not find that hippocampal replay had any differential relationship with memory for shared vs. unique features of the satellites (nor for verbal vs. visual features). This result is consistent with the observation in this study and in our previous study[30] that there is no interaction in memory change across sessions between Sleep vs. Wake group and unique vs. shared features. This suggests that there is coherent replay of the entire object representation that does not favor particular properties. While both feature types appear to benefit equally from replay and from sleep, it may be, as mentioned above, that systems consolidation during sleep allows shared features to undergo and maintain a memory boost above the pre-sleep baseline. The lack of specificity in the relationship between replay and memory for particular feature types also suggests that these results are likely to generalize to other forms of item memory. Broadly, we expect our findings to generalize to any two-test paradigm where replay is measured shortly after the initial test and shortly before the delayed test.

Our findings suggest that wake replay in the hippocampus does not simply reflect the strongest representations rising to the surface, but instead is adaptive, prioritizing memories that most need strengthening. The results provide evidence for hippocampal replay of individual memories in humans, and add to a growing literature suggesting that wake replay is associated with improved subsequent memory. The findings also point to a relationship between wake replay and the memory processing that occurs over a night of sleep, a promising dynamic to explore in future research.

## Methods

**Participants**. Twenty-four participants (14 females, mean age = 24.5 years, range = 19–38 years) from the Princeton University community participated in exchange for monetary compensation or course credit. Data from six additional subjects were excluded due to excessive motion (three subjects), technical issues (two subjects), and poor performance on the one-back task in the scanner (one subject; A′ >2 SD below average). Informed consent was obtained from all subjects, and the study protocol was approved by the Institutional Review Board for Human Subjects at Princeton University.

Subjects reported no history of neurological disorders, psychiatric disorders, major medical issues, or use of medication known to interfere with sleep. Subjects also reported having a regular sleep–wake schedule, which was defined as regularly going to bed no later than 2 am, waking up no later than 10 am, and getting at least 7 h of total sleep per night on average. The Epworth Sleepiness Scale[52] (ESS), the Morningness-Eveningness Questionnaire[53] (MEQ), and the Pittsburgh Sleep Quality Index[54] (PSQI) were used to screen out potential subjects with excessive

daytime sleepiness (ESS score >10), extreme chronotypes (MEQ <31 or >69), and poor sleep quality (PSQI >5).

**Stimuli.** Participants learned the features of 15 novel "satellite" objects organized into three "classes" (Fig. 1a). Each satellite has a class name (Alpha, Beta, or Gamma) shared with other members of the same category, a unique "code" name, and five visual parts. One of the learned satellites in each category is the prototype (shown on the left for each category in Fig. 1a), which contains all the prototypical parts for that category. Each of the other satellites has one part deviating from the prototype. Thus, each non-prototype shares four features with the prototype and three features with other non-prototypes from the same category. Exemplars from different categories do not share any features. Satellites were constructed randomly for each participant, constrained by this category structure.

**Procedure prior to experiment.** Subjects were instructed to attempt at least 7 h of sleep per night for 5 nights prior to (and during, in the case of Sleep subjects) the study. Adherence to the sleep schedule was tracked with daily sleep diaries. Subjects were also asked to refrain from any alcohol for 24 h prior to the first session, and throughout the experiment, and to keep caffeine intake to a minimum during this period. Heavy caffeine users (>3 servings per day) were not enrolled.

**Procedure for Session 1 training.** Participants learned about the satellites in two phases. In the first phase, which lasted 15 min on average, satellites were introduced one by one, with each of the 15 satellites shown once. For each satellite, the class and code name were displayed, followed by the image of the satellite. A box highlighted each of the five visual features on the satellite image one by one, to encourage participants to attend to each feature. Participants were then asked to recall the class and code names by clicking on one of three options given for each name. Next, participants used a point-and-click interface to try to reconstruct the satellite image. Icons representing the five part types were displayed on the right-hand side of the screen, and when an icon was clicked, all the possible versions of that part were displayed in a row on the bottom of the screen. The participant could then click on one of the part versions on the bottom to add it to the satellite in the center of the screen. If the participant was too slow at this task (took longer than 15 s), or reconstructed the satellite incorrectly, a feedback screen would appear displaying the correct features.

In the second phase of training, which lasted 32 min on average, participants were shown a satellite with one feature missing, which could be one of the five visual features, the code name, or the class name (code and class name buttons were displayed along with the part icons on the right hand side of the screen, and when selected, displayed the corresponding name options in a row on the bottom). Using the same point-and-click interface, participants chose a feature (out of all possible) to complete the satellite. If they chose the correct feature, they were told it was correct, and could move on to the next trial. If they chose an incorrect feature, they were shown the correct feature, and had to repeat the trial until they chose the correct feature.

Remembering the shared properties of the satellites is easier than remembering the unique properties, as the shared properties are reinforced across study of all the satellites in the same category. The task was titrated in pilot testing to ensure that, at the end of training, participants performed equivalently at retrieving shared and unique properties of the satellites. To accomplish this, unique features were queried 24 times more frequently than shared features. This phase of training continued until the participant reached a criterion of 66% of trials correct on a block of 32 trials, or until 60 min had passed. Only one participant did not reach the criterion, but was very close, with an accuracy of 63%.

**Procedure for Session 1 test.** Immediately after training, participants were tested by again filling in missing features of the satellites, now without feedback. The test phase had 39 trials, with two missing features per trial, which allowed us to collect more information per trial as well as provide less exposure to the correct features (to minimize learning during the test phase). The test phase took 10 min on average. Each satellite appeared twice in the test phase: once with its code name and its class name or one shared part tested, and once with two shared parts or one unique part and one shared part tested. The remaining nine trials tested generalization to novel satellites. Novel satellites were members of the trained categories but had one novel feature. The queried feature for novel items was always a shared part (class name or shared visual feature). Test trials were presented in a random order.

**Procedure for fMRI scanning.** After completion of the first session test phase, participants were scanned while viewing the satellite images (without names) for 52 min. Satellites subtended up to 19° of visual angle on the scanner projection. There were eight runs, lasting 6.5 min each, with self-paced breaks between runs. In each run, each of the 15 images was presented four times in pseudo-random order, such that each satellite appeared in the first, second, third, and fourth quarter of the run. Four trials in each run were randomly chosen to be duplicated, such that these satellites were shown twice in immediate succession. These served as rare (four trials out of 64) targets for a one-back task that subjects performed while viewing the satellites, to encourage maintenance of attention. Subjects pressed one key on a

keypad to indicate that the current satellite was not an exact repetition of the previous satellite, and a different key to indicate that it was a repetition. Keys corresponded to index and middle fingers of the right hand, with key assignment for repetition and no-repetition counterbalanced across subjects. Feedback for responses at each trial was provided as a green or red dot at fixation. Each satellite was presented for 3 s with a jittered interstimulus interval (40% 1 s, 40% 3 s, 20% 5 s) to facilitate modeling of the response to individual items.

Next, we collected a ninth 6.5-min functional run where participants were instructed to relax and watch the fixation dot on the screen, emphasizing that they should keep their eyes open.

**Procedure for Session 2.** In Session 2, participants did the same scan procedure (52 min of satellite viewing followed by 6.5 min eyes open rest), with images presented in a different random order. Then they got out of the scanner and completed the same test phase as in the first session, with identical trials presented in a different random order. The KSS, which assesses state sleepiness/alertness on a scale of 1 (extremely alert) to 9 (very sleepy), was completed at the beginning and end of each session.

Participants in the Sleep group ($n = 12$) began the first session around 7 pm and the second session around 9 am, and participants in the Wake group ($n = 12$) began the first session around 9:30 am and the second session around 10 pm (time choices were constrained by scanner availability). They were not told in the first session or at the beginning of the second session that there would be a second memory test, though when asked afterwards, they generally reported that they were not surprised by it. Only two participants reported trying to think about the satellites between the two tests. Subjects in the Wake condition were instructed not to nap between sessions.

**fMRI data acquisition.** Data were acquired using a 3T Siemens Skyra scanner with a volume head coil. In each session, we collected nine functional runs with a T2\*-weighted gradient-echo EPI sequence (36 oblique axial slices: $3 \times 3$ mm inplane, 3 mm thickness; echo time = 30 ms; repetition time (TR) = 2000 ms; flip angle = 71°; matrix = $64 \times 64$). Each run contained 195 volumes. We collected two anatomical runs for registration across subjects to standard space: a coplanar T1-weighted FLASH sequence and a high-resolution 3D T1-weighted MPRAGE sequence. An in-plane magnetic field map image was also acquired for EPI undistortion.

**Regions of interest (ROIs).** The hippocampus ROIs were calculated for each participant in subject space using automatic segmentation in Freesurfer[55]. ROIs of hippocampus subfields CA1 and CA2/3/DG were defined from a probabilistic atlas of the medial temporal lobe[56], projected into subject space for analyses.

**fMRI preprocessing.** Functional runs were preprocessed using FMRI Expert Analysis Tool (FEAT) Version 5.98, part of FSL (FMRIB's Software Library, www.fmrib.ox.ac.uk/fsl), including: removal of first four volumes, motion correction using MCFLIRT (individual runs with excessive estimated motion were excluded: 31 out of 384 runs); fieldmap-based EPI unwarping using PRELUDE + FUGUE; slice-timing correction using Fourier-space time-series phase-shifting; non-brain removal using BET; spatial smoothing using a Gaussian kernel of FWHM 5 mm; and high-pass temporal filtering using a 64s-sigma Gaussian kernel. Functional runs were registered with FLIRT to the FLASH image, the MPRAGE image, and an MNI standard brain with interpolation to 2 mm isotropic voxels.

**General Linear Model (GLM) for satellite templates.** We modeled the evoked response to individual satellites across a dataset concatenating the eight pre-processed runs for satellite-viewing periods. The GLM was fit using FILM with local autocorrelation correction in FSL. The model contained a regressor for each of the 15 satellites. Each regressor had a delta function for every 3 s presentation of the satellite, excluding repetitions added for the one-back task, convolved with a double-gamma hemodynamic response function. There were also eight regressors indicating which run the data at that TR corresponded to. The resulting parameter estimates reflected the response of all voxels to each individual satellite, which we call the satellite's template. For each participant, two sets of templates were calculated, one for each session.

We ran control analyses where we scrambled the satellite labels before running the GLM. After removing the second item in back-to-back item repetitions (used for the one-back task), the order of the images was randomized such that each image was seen the same number of times (relative to before scrambling) but likely no longer in the correct position. The GLMs and replay analyses were then carried out using these incorrect templates. These analyses were done to ensure that any replay detected or relationship with behavior was not an artifact of the analysis pipeline but instead reflected reinstatement of the specific satellite representations.

**Replay analysis.** Within each session for each participant, we compared each of the estimated templates to the pattern of activity at each TR of the rest period to find potential replay events. After preprocessing the data from the rest period as described above, the time series was low-pass filtered through a convolution with a three-point-width Hamming window. This transformation makes the data

frequency closer to that of the hemodynamic response function, serving as a replacement for event-related convolution in this scenario where event timing is unknown. We next converted to percent signal change and excluded any voxels with variance more than three SDs from the mean. Then, for all voxels within an ROI or a searchlight (defined as described below), we computed the Pearson correlation between each template from a given session and the pattern of activity at each TR of the rest period from that session (Fig. 3a). This resulted in a matrix of positive and negative values for 15 satellites by 191 TRs in the rest period (Fig. 3b, left).

To narrow this matrix down to potential replay events, we only considered activity with strong correspondence between templates and rest period activity. At each TR, we defined strong correspondence as more than 1.5 SDs above the mean across the 15 satellites (1.5 value based on a prior fMRI replay study[7]). We chose this approach to increase our chances of seeing satellite-specific replay, as opposed to a brain state at a given TR that resembles satellites in general, while still allowing for the possibility that multiple satellites (or none) are replayed at a given TR.

After thresholding the matrix at each TR (Fig. 3b, right), we summed the total amount of potential replay activity across all TRs for each satellite. We then computed the Pearson correlation between the sum for each satellite and an individual's test accuracy for each satellite, where accuracy is the average performance across the four features queried (Fig. 3c). Across-subject statistics were calculated on these Fisher-transformed correlation values.

We ran two variants on this pipeline as control analyses to assess the degree to which replay was item specific. The first used the 1.5 SD threshold only across satellites within the same category, such that each satellite's replay was evaluated relative to the evidence of replay only of other satellites in the same category. The second analysis shuffled the labels of satellites from the same category when calculating the template-rest period correlations (Fig. 3a). Each satellite was randomly assigned a different category member's template for a particular rest period, and then correlations with behavior were calculated for the original satellite. This analysis was done four times, rotating each item through each of the other four category member's templates, and the results were averaged across rotations.

**Representation of category structure**. Within the hippocampus and subfield ROIs, we calculated all pairwise correlations between templates corresponding to satellites from the same category and all correlations between templates for satellites from different categories, and subtracted the average between-category correlation from the average within-category correlation. This provides an estimate of the extent to which objects from the same category are represented similarly.

**Searchlight analyses**. We additionally ran the replay and representation-of-category-structure analyses within every $3 \times 3 \times 3$ voxel cube in the brain. We assigned the final replay–behavior correlation value (Fig. 3c) for the replay analysis, or the correlation difference value for the category-structure analysis, to the center voxel of each searchlight, and then projected these maps to MNI space to allow comparison across participants. We used the *randomise* function in FSL to perform permutation tests for reliable clusters, using a cluster formation threshold corresponding to $p = 0.01$.

**GLM for univariate change effects**. We also ran a GLM to test for overall univariate differences in activity level between the first and second session. Instead of one regressor for each satellite, the model had one regressor with a delta function at every image presentation. We subtracted the map for Session 1 from the map for Session 2 and assessed cluster reliability for Sleep and Wake groups separately as well as contrasting Sleep and Wake groups.

## Data availability

The MRI and behavioral test data that support the findings of this study are available at OpenNeuro.org under accession code ds001454 (https://openneuro.org/datasets/ds001454, https://doi.org/10.18112/openneuro.ds001454.v1.3.0).

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

## Acknowledgements

We thank Luis Piloto for help with initial analyses and subject running, Michael Arcaro for advice on processing the rest data and identification of visual areas, and Lila Davachi and Nancy Kanwisher for useful analysis suggestions. This work was supported by the National Institutes of Health (NINDS F32-NS093901 to A.C.S., NIA R01-AG046646 to S. C.M., and NIMH R01-MH069456 to K.A.N.) and the National Science Foundation (GRFP to E.A.M. and BCS-1439210 to S.C.M.).

## Author contributions

A.C.S., E.A.M., T.T.R., S.C.M., and K.A.N. contributed to designing the study. A.C.S. collected and analyzed the data in consultation with K.A.N. All authors discussed the interpretation of the data. A.C.S. drafted the manuscript and all authors edited the manuscript.

## Additional information

**Competing interests:** The authors declare no competing interests.

