## [Peer Review File · Nature Communications]

Reviewers' Comments:

Reviewer #1:

Remarks to the Author:

Schapiro and colleagues investigated how 'putative' replay of memory traces in the hippocampus post-encoding predict memory, as well as the putative role of sleep. The authors show that immediately after memory encoding and a subsequent memory test, reactivation was greatest for items in which an individual had the worst memory performance. However, prior to a subsequent delayed memory test, in which a memory test was expected, reactivation was positively correlated with performance. This paper is strengthened by methodological innovation, that provide unique characterizations of post-encoding and pre-retrieval processes during rest. However, there are major weaknesses in the paper ranging from study design, analysis approaches, and theoretical framings that derail from the overall impact. Notably, I have previously reviewed this manuscript, and the review will be similar to prior reviews as many concerns were unaddressed.

Comment 1: There is a very strong confound in the experimental design, that makes the interpretation of the data quite difficult. Post-encoding processes occur after the memory test in session 1, and prior to the memory test on session 2. Differences in brain-behavior relationships could then either be related to how far away from encoding consolidation is being measured, which is closer to the authors' interpretation, or alternatively whether rest was before or after memory retrieval. For example, the negative relationship between memory performance and rest during session 1 could reflect post-retrieval monitoring of bad memory performance whereas, session 2 could reflect active rehearsal prior to an upcoming memory test. This problem is especially relevant as participants expected the memory test in session 2. This design makes it hard to understand which domains of memory in which the findings generalize.

Comment 2: Another major weakness of the manuscript is the complexity of the learning session and memory test. Learning consists of multiple phases, a mixture of fixed learning versus learning to criteria, a hierarchical/categorical structure of items to-be-learned, and different numbers of exposure to different feature types. Similarly, the memory tests test multiple features of the item that may rely on different retrieval processes (i.e. pattern separation versus completion). The authors then use a relatively simplistic summary statistic of memory. By glossing over the layers of complexity, their brain-behavioral analysis are over-simplified. I am further concerned about how generalizable their findings are to learning in other contexts. Given that many other studies show replay contributes to better memory, it may be that only one of the factors listed above are contributing to their unique pattern of findings.

Comment 3: I would need additional analysis to support the authors' claims that their findings relate to item-specific reactivation versus category-specific activation. In the current analyses set, they code the data as whether it differs from the other 15 items (which a majority of are from a separate category). Do the authors' findings replicate if they tag their items as 'replay' by looking at differences within category during rest versus across category? These positive results are warranted to support the authors' claims.

Comment 4: The authors mainly identify pattern/reinstatement differences in the MTL, but see univariate activation in the PFC. More discussion is warranted on the nature of these two types of signals, and how they inform theoretical implications of systems consolidation.

Comment 5: A few studies have shown that item-specific reinstatement outside of the hippocampus predict positive memory strength. Could the authors describe their findings in relationship to this prior work. Do they think differences emerge due to region (i.e., hippocampus versus cortical MTL) or due to task structure (i.e., following and/or preceding an expected memory test).

Comment 6: I find the use of "replay" throughout the manuscript to be problematic. There is a

very precise definition of what replay means within the rodent literature, that refers to sequential reactivation of events, I believe the use of reactivation is more appropriate, and claims of identifying hippocampal replay may be over-selling the current findings.

Reviewer #2:

Remarks to the Author:

This study investigated item-level replay in human subjects using fMRI pattern analysis. Subjects learned features of different objects in the fMRI scanner and hippocampal replay was assessed twice during awake rest periods, once immediately after learning and a second time ~12 hours later after a night of sleep or a day of wakefulness. The results showed that i) weakly-learned information was replayed more during the first session after learning, ii) replay during the second session predicted memory performance during subsequent test, and iii) replay in both sessions predicted memory improvement from learning to test only in subjects who slept during the retention interval.

These findings are novel and highly interesting in advancing our understanding of hippocampal replay during awake periods in humans. The methods appear to be sound and the manuscript is well-written. I only have a few comments that should be addressed.

1. My main concern is with the main analyses for effects of group, hemisphere and feature type (on page 13): for each variable data were collapsed across all other variables, which results in the main effects for each variable. I do follow the authors' reasoning to do this. However, in this way possible interactions between single factors remain unclear. Therefore, it is not possible to draw the conclusion that none of the factors had an effect because the factors in combination (i.e. their interaction) might have an effect. At the very least this needs to be discussed and the conclusions need to be adapted accordingly.
2. For the cross-session replay-behavior relationships, I would be interested if there were any differences for the sleep and wake groups. Especially for the correlation between session 1 replay and session 2 behavior (regressing out session 1 behavior), the analyses should be reported separately for the sleep and wake groups, considering that wake replay in session 1 might trigger processes that continue during sleep and affect subsequent memory performance.
3. With regard to the experimental task, some details should be clarified. On page 5, it is mentioned that "participants learned about 15 novel "satellite" objects" – what does "about" mean here? Was it not exactly 15 for all subjects, i.e. five of each class? Also, did the learning satellites include the prototype? When subjects were required to "reconstruct the satellite images from scratch" during the training session, was this done for all of the 15 learned satellites?
4. In the methods section, it says that the training session continued until the learning criterion of 66% correct trials was reached or until 60 minutes had passed. How many subjects did not reach the criterion within the 60 minutes? Do the results differ for these subjects?

Reviewer #3:

Remarks to the Author:

Schapiro and colleagues present compelling new fMRI data suggesting how memories might be selected for replay in humans. Participants were trained to identify different "classes" of satellites, which had both shared (common to all satellite exemplars within a class) and unique (specific to one particular satellite exemplar) features. Memory and neural replay of specific satellites were both measured during the first session, and again during session a second session ~12 hours later. Half of the participants slept between sessions (session 1 in PM, session 2 in AM), and the other half did not (session 1 in AM, session 2 in PM). The main findings were that (1) the sleep group showed less forgetting of unique visual features and improved memory for the shared visual features as compared with the wake group, (2) more replay was observed in hippocampus for

items remembered less well during session 1 and items remembered better during session 2, and (3) replay of specific items was associated with memory improvement (S2-S1 performance) only in the sleep group. These findings suggest that (among other things) weak memories receive preferential replay during awake rest periods, and that the effects of awake replay may be enhanced when a delay includes sleep.

These are very exciting data from a carefully conducted study. Of note, all of the replay analyses are on an item-by-item basis, such that the authors are able to make claims about which specific memories (within a given participant) show evidence of replay. This feature makes the results especially timely and interesting, and I believe they will be widely influential in the field—memory replay is a hot topic, and this study fills a critical gap in describing one factor (memory strength) that dictates what is deserving of replay, and what is not. The statistical analyses are appropriate and valid. I am generally very enthusiastic about this paper; however, I do have a few questions and concerns:

1) I am left wondering the extent to which the order of replay versus test might determine the direction of the relationship between memory and replay. My intuition is that the impact of test on replay (first session) and the impact of replay on test (second session) could be the result of very different neurobiological mechanisms, and that might be part of the reason for the change in direction from session 1 to 2. For example, it could be the case that at the session 1 test, satellites on which the participant is incorrect are tagged for later replay (it seems to me that this could happen despite no external feedback). The authors address this possibility in the discussion. For session 2, it could be the case that satellites that are replayed are primed, and thus easier to access during the subsequent test. The item representation or “template” phase also serves as a reminder that could influence this process, which I assume is by design. Could the authors touch on these possibilities? (Related to this point, I quite liked the inclusion of the attention univariate control analysis, which seems to suggest participants aren’t simply paying attention to/studying those less familiar or “weaker” satellites more during the template phase.)

2) I would be interested to hear the authors’ thoughts on where these “weak” memories might lie on the continuum of strength in terms of everyday memory. The authors find that the weaker of the satellite memories showed preferential replay, but it strikes me as a key point that this “weak” is relative to the other satellites—all of which were more well-learned than many real world, one-off experiences. I presume there is a lower limit on this selection process—i.e., memories that are too weak might not be replayed at all (or may be tagged but too weak to be reactivated), whereas medium strength memories could be the prime target. It might be helpful, especially for a casual reader who does not dig into the methods of the present paper, to describe a bit about what is meant by “weak” in the grand scheme of things.

3) I believe that through the end of session 1, the experience of participants in the two groups was identical (with the exception of time of day). If I am correct, I believe any (numerical, even if not reliable) differences during session 1 (e.g., Figure 4B) are not meaningful. I apologize if this was in the paper and I missed it, but it would be useful to emphasize in the paper that for analyses restricted to session 1, no differences between groups would be expected.

4) Eyeballing the charts, it looks like the effects might right-lateralized in the wake group, and more bilateral in the sleep group. Was this the case, and if so, is that meaningful?

5) Given the resolution of the functional data (3x3x3mm), subfield analyses are not interpretable. In my opinion, these should not be performed at all at this resolution; if the authors wish to keep them in, I would suggest adding extensive caveats throughout the paper.

6) On p. 16, the authors write: “One study in rodents found that replay during rest was important for gradually-strengthened but not quickly-formed memories [6] ... , perhaps reflecting similar processes.” It is not clear to me from this wording why “gradually strengthened” memories would

be weaker and replayed more. Could the authors expand on this point or reword?

7) Clarification point: Were the same features tested for a given satellite for session 1 and session 2 (i.e., were the test trials identical with the exception of the trial order)?

Reviewer #1:

Schapiro and colleagues investigated how ‘putative’ replay of memory traces in the hippocampus post-encoding predict memory, as well as the putative role of sleep. The authors show that immediately after memory encoding and a subsequent memory test, reactivation was greatest for items in which an individuals had the worst memory performance. However, prior to a subsequent delayed memory test, in which a memory test was expected, reactivation was positively correlated with performance. There paper is strengthened by methodological innovation, that provide unique characterizations of post-encoding and pre-retrieval processes during rest. However, there are major weaknesses in the paper ranging from study design, analysis approaches, and theoretical framings that derail from the overall impact. Notably, I have previously reviewed this manuscript, and the review will be similar to prior reviews as many concerns were unaddressed.

We appreciate the Reviewer’s very helpful comments and hope they agree that our revisions have addressed their concerns and strengthened the paper.

Comment 1: There is a very strong confound in the experimental design, that makes the interpretation of the data quite difficult. Post-encoding processes occur after the memory test in session 1, and prior to the memory test on session 2. Differences in brain-behavior relationships could then either be related to how far away from encoding consolidation is being measured, which is closer to the authors interpretation, or alternatively whether rest was before or after memory retrieval. For example, the negative relationship between memory performance and rest during session 1 could reflect post-retrieval monitoring of bad memory performance whereas, session 2 could reflect active rehearsal prior to an upcoming memory test. This problem is especially relevant as participants expected the memory test in session 2. This design makes it hard to understand which domains of memory in which the findings generalize.

We understand the Reviewer’s point, and we realize we were not clear enough about the manipulation. We never intended this study to be a “pure” manipulation of time-since-encoding. Rather, our goal was to take a common learning scenario (with an immediate test and a delayed test) and “check in” on replay early and late in this interval between tests, to see how this replay relates to test performance. Thus the differential ordering of the memory test and the rest period in Sessions 1 and 2 is not a confound – rather, it is a necessary feature of our design: We wanted both rest periods to reside within the interval between tests, but placed at different points in this interval (shortly after the initial test & shortly before the delayed test).

Having said this, we think the Reviewer raises good points about the *mechanisms* that are responsible for the observed effects; most pertinently, to what extent are the “replay” processes that we are measuring deliberate/explicit vs. automatic/implicit? We had added a sentence to the discussion after the Reviewer brought this concern up in their comments at the previous journal:

“This replay seems likely to reflect incidental, endogenous activity, as participants did not report trying to rehearse the satellites and were not told that there would be a second memory test.”

Although participants were not told about the second memory test, they may have guessed that memory would be tested later. But even if they did expect a second test, they did not know when it might happen, and, most importantly, they did not report trying to rehearse the satellites in either rest period. It is still possible that participants were doing active rehearsal that they later forgot about. We have now added this possibility as a caveat in the discussion:

“These reports were collected at the end of the experiment, so it is possible that participants did do some strategic rehearsal that they later forgot. Future work could assess the degree of implicitness of the replay more directly by asking participants for reports throughout the experiment.”

Regardless of the degree of implicitness of the replay, it is worth asking whether the possibility of a future memory test was driving replay. It seems plausible that some estimation of future relevance of this information (as in a future memory test) would modulate replay. We think this is an important question for future work, and have added text to acknowledge this point:

“Another question for future work is whether the mere possibility of future memory tests was a driver of either implicit or explicit replay, which could be assessed in a design that leads participants to believe that their memory will not be tested again.”

To emphasize to the reader that the “flipped” ordering of the memory test and rest periods is a key feature of our design, we now start the section in the Results section on the Session 1 correlations: “In Session 1, the memory test came before the rest period.” And for Session 2: “In Session 2, the rest period came before the memory test.”

Because the memory test came before the rest period in Session 1, a remaining ambiguity is whether weaker memories are being tagged for further processing during the memory test, or whether that had already happened earlier during training. This is an important question for future work and we do not believe the ambiguity undermines the value of the current study. We now mention this in two places in the Discussion:

“Because the memory test always came between the training and rest periods in Session 1, we do not know whether the prioritization process identified weak memories during the training phase or the test phase. Future studies could probe this by having a rest period immediately after the training phase.”

“One hypothesis suggested in other work [33, 34] is that processing during wake rest serves to “tag” memory traces for subsequent replay during sleep. Alternatively, it may be that some other factor occurring prior to rest (either during learning, or during the memory test) drives prioritization during both wake and sleep replay...”

Altogether, we greatly appreciate the reviewer pushing us to consider these alternative explanations and to explicitly discuss them in the text. With regard to the reviewer’s question about the generality of our results: The learning scenario explored here (with an immediate and a delayed test) is quite common, and we expect that our results will generalize to any two-test paradigm where replay is measured shortly after the initial test and shortly before the delayed test (see also our response to Comment 2 below, for additional points regarding the generality and robustness of our results). We now say this in the Discussion:

“Broadly, we expect our findings to generalize to any two-test paradigm where replay is measured shortly after the initial test and shortly before the delayed test.”

Comment 2: Another major weakness of the manuscript is the complexity of the learning session and memory test. Learning consists of multiple phases, a mixture of fixed learning versus learning to criteria, a hierarchical/categorical structure of items to-be-learned, and different numbers of exposure to different feature types. Similarly, the memory tests test multiple features

of the item that may rely on different retrieval processes (i.e. pattern separation versus completion). The authors then use a relatively simplistic summary statistic of memory. By glossing over the layers of complexity, their brain-behavioral analysis are over-simplified. I am further concerned about how generalizable their findings are to learning in other contexts. Given that many other studies show replay contributes to better memory, it may be that only one of the factors listed above are contributing to their unique pattern of findings.

We used this paradigm because we had developed it for our previous behavioral study (Schapiro et al., 2017, *Scientific Reports*) and found that there were robust effects of sleep on memory for these objects, with interesting differences between feature types. We actually simplified that design for this experiment, removing a manipulation of the frequency of presentation of the different categories during training. The task is perhaps unusual or complicated from the perspective of the episodic memory literature, but it is quite standard as an assessment of category learning. Introduction of objects followed by property inference until a learning criterion is met is a standard design (e.g., Chin-Parker & Ross, 2004, Diagnosticity and Prototypicality in Category Learning: A Comparison of Inference Learning and Classification Learning, *JEP: LMC*). We decided to use this task instead of a classic episodic memory task because we were interested in the possibility that replay might differentially affect memory for the different aspects of the objects.

The Reviewer wonders whether our results may be driven by particular aspects of this training process or particular aspects of the object memory. Future work will be needed to assess whether the results are dependent on the details of the training process, but we were able to assess the impact of different aspects of the object memory. We show that the results were the same whether assessing memory for features unique to an individual satellite or shared across members of a category (Figure 4c). Similarly, the results were the same whether the feature was a name or visual property (Figure 4d). These results suggest that replay is acting in the same way on all aspects of the object memory, which provides some evidence that these results are likely to generalize to other kinds of memory. We thus view the assessment of multiple feature types as a feature rather than a bug, and we now highlight this point:

“We did not find that hippocampal replay had any differential relationship with memory for shared vs. unique features of the satellites, nor for verbal vs. visual features. This suggests that there is a coherent replay of the entire object representation that does not favor particular properties. The lack of specificity in these relationships also suggests that these results are likely to generalize to other forms of item memory.”

Comment 3: I would need additional analysis to support the author’ claims that their findings relate to item-specific reactivation versus category-specific activation. In the current analyses set, they code the data as whether it differs from the other 15 items (which a majority of are from a separate category). Do the authors findings replicate if they tag their items as ‘replay’ by looking at differences within category during rest versus across category? These positive results are warranted to support the authors’ claims.

We appreciate the Reviewer prompting us to run additional analyses along these lines, as we agree they would strengthen the paper. We implemented several different analyses to address this issue of whether we are looking at item-specific, within-category replay. The results of all analyses support the conclusion that the replay we are observing is indeed item specific.

First, we note that we added an analysis to the originally submitted version of the paper for Nature Communications based on similar comments the Reviewer made in their review of the

paper for a previous journal.

In the Methods:

“We ran control analyses where we scrambled the satellite labels before running the GLM. After removing the second item in back-to-back item repetitions (used for the one-back task), the order of the images was randomized such that each image was seen the same number of times (relative to before scrambling) but likely no longer in the correct position. The GLMs and replay analyses were then carried out using these incorrect templates. These analyses were done to ensure that any replay detected or relationship with behavior was not an artifact of the analysis pipeline but instead reflected reinstatement of the specific satellite representations.”

In the Results for Session 1:

“We ran the same analysis after scrambling the labels of the satellites during the estimation of the template representations. The scrambling procedure should render the templates meaningless and result in less replay detection and no relationship with behavior. Indeed, after scrambling, there was significantly less detected replay overall ($t[23]=2.520$, $p=0.019$) and no relationship with behavior (mean=-0.020, $t[23]=0.404$, $p=0.690$). This suggests that the measured replay activity in the analyses above reflects the reinstated representation of specific satellites.”

In the Results for Session 2:

“We again ran this analysis after scrambling the labels of the satellites during the estimation of the template representations. After scrambling, there was again significantly less detected replay overall ($t[23]=2.790$, $p=0.010$) and no relationship with behavior (mean=0.0005, $t[23]=0.011$, $p=0.991$).”

While we believe this analysis helps bolster the claim that the replay is item-specific, it does not directly address the idea of within-category specificity. To do this, we ran three new analyses. First, we changed the 1.5 SD criterion for replay to apply across the 5 satellites within the same category, as opposed to across all 15 satellites (as suggested by the Reviewer in their previous review). This requires that an item emerge more strongly than other members of its category to count as being replayed. The results were qualitatively very similar using this within-category thresholding. Mean correlation between replay and memory in Session 1 in left hippocampus was -0.124 ($t[23]=2.003$, $p=0.057$) and in right hippocampus was -0.086 ($t[23]=1.399$, $p=0.175$). In Session 2, the mean correlation between replay and memory in left hippocampus was -0.027 ($t[23]=0.332$, $p=0.743$) and in right hippocampus was 0.184 ($t[23]=2.814$, $p=0.010$). The difference between the correlations in Session 1 vs. 2 was reliable in right hippocampus ($t[23]=3.398$, $p=0.003$; left hippocampus: $t[23]=0.874$, $p=0.391$).

In a second analysis, instead of running correlations between the replay sums for the 15 satellites and the memory scores for those 15 satellites, we averaged the replay sums and behavioral scores for satellites of the same category, and computed correlations across the resulting three values for each subject. To the extent that the neural responses carry information about the category of the satellite and not individual satellites, the results of this analysis should look similar to the main analysis. However, the effects were completely different: In Session 1, the correlation between replay and memory in left hippocampus was -0.086 and in right hippocampus was -0.036. In Session 2, the correlations were -0.470 and -0.049 in left and right hippocampus, respectively. No individual correlations or differences between Session 1 and 2 were significant ($ps>0.122$). These results provide some indication that the main analyses were uncovering item-specific

information. However, the analysis may be noisy or unstable, since correlations were calculated across only three data points for each subject.

We therefore ran a third analysis which avoids this issue. In this analysis, when calculating the similarity between satellite templates and the pattern of activity arising at every TR in the rest period, we swapped template patterns for satellites within the same category. Each satellite was randomly assigned a different category member's template for a particular rest period, and then correlations with behavior were calculated for the original satellite. Again, to the extent that the neural responses carry information about the category of the satellite and not individual satellites, the results of this analysis should look similar to the main analysis. The results were again qualitatively different (Session 1: mean=0.009, $t[23]=0.440$, $p=0.664$; Session 2: mean=-0.034, $t[23]=1.674$, $p=0.108$). The Session 1 correlations were significantly higher than in the correctly-aligned data ($t[23]=2.926$, $p=0.008$) and the Session 2 correlations were significantly lower ($t[23]=4.179$, $p=0.0004$). These findings provide strong evidence that the templates are matching up with the neural representations of specific items within a category.

We have now added the first and third analyses to the paper (the second analysis may not be very informative given the small number of data points used to compute correlations, though we thought the Reviewer might be interested to see it). In the Results section, we have created a new section called *Specificity of the replay to individual satellites* to consolidate these new analyses together with the scrambling analysis already included.

In the Methods:

“We ran two variants on this pipeline as control analyses to assess the degree to which replay was item specific. The first used the 1.5 SD threshold only across satellites within the same category, such that each satellite's replay was evaluated relative to the evidence of replay only of other satellites in the same category. The second analysis shuffled the labels of satellites from the same category when calculating the template-rest period correlations (Fig. 2a). Each satellite was randomly assigned a different category member's template for a particular rest period, and then correlations with behavior were calculated for the original satellite. This analysis was done four times, rotating each item through each of the other four category member's templates, and the results were averaged across rotations.”

In the Results:

“*Specificity of the replay to individual satellites.* We ran the same replay analyses after scrambling the labels of the satellites during the estimation of the template representations. The scrambling procedure should render the templates meaningless and result in less replay detection and no relationship with behavior. Indeed, after scrambling, there was significantly less detected replay overall (Session 1: $t[23]=2.520$, $p=0.019$; Session 2: $t[23]=2.790$, $p=0.010$, collapsed across right and left hippocampus) and no relationship with behavior (Session 1: mean=-0.020, $t[23]=0.404$, $p=0.690$; Session 2: mean=0.0005, $t[23]=0.011$, $p=0.991$). This suggests that the measured replay activity in the analyses above reflects the reinstated representation of specific satellites. To further assess whether we were observing replay of individual satellites as opposed to a more abstract representation of a satellite's category, we ran two additional control analyses. One analysis shuffled the labels of satellites from the same category during the calculation of template-rest period correlations, and again resulted in no evidence of meaningful replay-behavior relationships (Session 1: mean=0.009, $t[23]=0.440$, $p=0.664$; Session 2: mean=-0.034, $t[23]=1.674$, $p=0.108$). The Session 1 correlations were significantly higher than in the correctly-aligned data ($t[23]=2.926$, $p=0.008$) and the Session 2 correlations were significantly lower

($t[23]=4.179, p=0.0004$). These results provide strong evidence that the templates are matching up with the neural representations of specific items within a category. Finally, as a positive control, we assessed whether the results would remain the same if replay thresholds were calculated with respect to other items from the same category, as opposed to all possible items. The results were indeed qualitatively similar. The mean correlation between replay and memory in Session 1 in left hippocampus was -0.124 ($t[23]=2.003, p=0.057$) and in right hippocampus was -0.086 ($t[23]=1.399, p=0.175$). In Session 2, the mean correlation between replay and memory in left hippocampus was -0.027 ($t[23]=0.332, p=0.743$) and in right hippocampus was 0.184 ($t[23]=2.814, p=0.010$). The difference between the correlations in Session 1 and 2 was reliable in right hippocampus ($t[23]=3.398, p=0.003$; left hippocampus: $t[23]=0.874, p=0.391$). Overall, these analyses suggest that we are observing replay of individual satellite representations.”

In the Discussion:

“This demonstration of item-specific replay is grounded in the observed correlations (computed across-items, within-subjects) between replay and memory performance and bolstered by control analyses demonstrating that the replay is not generalized across items within the same category.”

Comment 4: The authors mainly identify pattern/reinstatement differences in the MTL, but see univariate activation in the PFC. More discussion is warranted on the nature of these two types of signals, and how they inform theoretical implications of systems consolidation.

We agree that this point deserves further discussion. We now include this in the Discussion section:

“These results may reflect an increased reliance on cortical areas as a result of systems consolidation [53], as cortical areas employ more overlapping representations than the hippocampus and should therefore facilitate memory for shared features [30, 54]. One cortical region often implicated as a consolidation site is the mPFC [55], and our whole-brain univariate analysis found increased activation across sessions in that region for the Sleep but not the Wake group. We did not find evidence for multivariate changes — in replay or representational similarity — in cortical regions, which may mean multivariate changes did not occur on this timescale, or it may simply be a null result due to methodological or power limitations. The univariate finding suggests that the mPFC is overall more engaged by the satellites after a period of sleep, but leaves open whether there are changes in its representational content.”

Comment 5: A few studies have shown that item-specific reinstatement outside of the hippocampus predict positive memory strength. Could the authors describe their findings in relationship to this prior work. Do they think differences emerge due to region (i.e., hippocampus versus cortical MTL) or due to task structure (i.e., following and/or preceding an expected memory test).

Our study and findings differ from previous human studies and findings in two main ways. One way is that previous studies always looked at replay preceding a memory test, whereas we look at replay both following and preceding a memory test. We replicate prior results in finding that replay (in our case, hippocampal) predicts positive memory strength, and we add the additional finding that poor memory predicts more replay. This additional finding helps in the interpretation of our and others’ findings that replay predicts memory, because it rules out the possibility that we just replay things that we already know better (as opposed to replay actually benefitting the memory.) The second way that our study differs is that we find replay in the hippocampus,

whereas prior studies hypothesized hippocampal involvement but did not find it. Evidence of hippocampal involvement from the rodent literature is overwhelming, so we think it is likely that the null findings in prior studies derive from methodological limitations.

In the Discussion, we say:

“In Session 2, items that were replayed more showed better subsequent memory. This result is consistent with accumulating findings from both the rodent [2, 5] and human literature [7-17, 52] suggesting a functional role for hippocampal replay (i.e., improved consolidation).”

“Prior fMRI studies showing a positive relationship between replay and subsequent memory [7-16] fit with the idea that replay causes better memory. However, these results can also be explained in terms of stronger memories being replayed more (without there necessarily being a causal relationship between replay and subsequent memory). Our results from the second session fit with this latter story, but the results from the first session do not.”

“While prior work has shown item-specific replay outside the hippocampus [7-9] and coarser replay within the hippocampus [10, 14, 46], the current results provide evidence for item-specific replay in the human hippocampus.”

Comment 6: I find the use of “replay” throughout the manuscript to be problematic. There is a very precise definition of what replay means within the rodent literature, that refers to sequential reactivation of events, I believe the use of reactivation is more appropriate, and claims of identifying hippocampal replay may be over-selling the current findings.

We are very sensitive to this issue and aware that there are differences of opinion about use of this word. We have considered switching to “reactivation” and have been polling our colleagues in the sleep and memory field (in person and on twitter) what they think of the word replay in this context. Some have taken the Reviewer’s position – that replay should be reserved specifically for sequential reactivation. Others (both rodent and human researchers) have argued that using the word replay is useful because it helps to create a bridge to the rodent literature and starts a conversation about whether these phenomena are related. We tend to agree with this latter position.

It also seems that rodent researchers have focused on sequential domains partly because it is easier to detect sequential replay, as opposed to a particular static pattern of activation, when recording from a relatively small number of neurons. In fMRI, we have the opposite methodological challenge – we have access to a large number of neurons, but relatively poor temporal resolution, making static reactivation of a pattern easier to detect than sequential reactivation (though the underlying activity may be meaningfully sequential in fMRI paradigms as well). We want to resist letting these methodological differences drive our nomenclature, to the extent that we are motivated in looking for reactivation by the same ideas and findings.

It may be that upon further study, it does turn out that reactivation of a set of features of an object is fundamentally different than reactivation of sequential experience, and at that point the nomenclature can be clarified. For now, we see value in putting forward the hypothesis that these phenomena are fundamentally related. In the previously submitted version of the paper, we had a footnote that read:

“Though the word “replay” is sometimes reserved for the sequential replay of place cell activity observed in rodents, we use the term more broadly to mean neural reactivation of recent

experience, including that observed in fMRI.”

We have now added another two sentences to further clarify our purpose in using the word:

“In using the same term as in the rodent literature, we posit that the reactivation we observe has fundamental properties in common with that observed in the rodent literature. Future work will determine whether these phenomena are indeed related enough to continue to deserve the same nomenclature.”

Reviewer #2 (Remarks to the Author):

This study investigated item-level replay in human subjects using fMRI pattern analysis. Subjects learned features of different objects in the fMRI scanner and hippocampal replay was assessed twice during awake rest periods, once immediately after learning and a second time ~12 hours later after a night of sleep or a day of wakefulness. The results showed that i) weakly-learned information was replayed more during the first session after learning, ii) replay during the second session predicted memory performance during subsequent test, and iii) replay in both sessions predicted memory improvement from learning to test only in subjects who slept during the retention interval.

These findings are novel and highly interesting in advancing our understanding of hippocampal replay during awake periods in humans. The methods appear to be sound and the manuscript is well-written. I only have a few comments that should be addressed.

We appreciate the Reviewer’s positive assessment of the manuscript and their careful read and useful analysis suggestions.

1. My main concern is with the main analyses for effects of group, hemisphere and feature type (on page 13): for each variable data were collapsed across all other variables, which results in the main effects for each variable. I do follow the authors’ reasoning to do this. However, in this way possible interactions between single factors remain unclear. Therefore, it is not possible to draw the conclusion that none of the factors had an effect because the factors in combination (i.e. their interaction) might have an effect. At the very least this needs to be discussed and the conclusions need to be adapted accordingly.

We appreciate the Reviewer’s point. The most straightforward analysis to assess interactions between these factors would have been an ANOVA, but that is not possible due to many missing cells in cases where subjects performed at floor or ceiling. However, mixed effects models are able to gracefully handle missing data, so we now include an assessment of interactions using mixed effects models. We also include a caveat that we likely do not have the power to confidentially explore interactions in this study, and that future work will be needed assess this.

“We also ran mixed effects models for each session with Sleep vs. Wake as across-subjects factor and verbal vs. visual, left vs. right hippocampus, and unique vs. shared as within-subjects factors. Mixed effects models can gracefully handle missing data, allowing us to assess potential interactions between factors. We did not find evidence for reliable interactions between any factors in Session 1. We did find evidence for an interaction between unique vs. shared and verbal vs. visual information in Session 2, with replay more positively associated with visual shared features than verbal shared features, and more positively associated with verbal unique features than visual unique features (X^2 for models with vs. without interaction = 5.70, $p=0.017$). These

analyses should be interpreted with caution, as power for assessing interactions in this study is low [42]. Potential interactions between these variables will need to be further assessed in future work.”

2. For the cross-session replay-behavior relationships, I would be interested if there were any differences for the sleep and wake groups. Especially for the correlation between session 1 replay and session 2 behavior (regressing out session 1 behavior), the analyses should be reported separately for the sleep and wake groups, considering that wake replay in session 1 might trigger processes that continue during sleep and affect subsequent memory performance.

This is a very interesting idea that we had not considered. We checked whether Session 1 replay is more related to Session 2 behavior (regressing out session 1 behavior) for the sleep group relative to the wake group, and found evidence to support this ($p=.039$, one-tailed, as this is a strong directional prediction). We now include this result:

“We also tested the idea that replay in Session 1 in the Sleep group might be more directly predictive of Session 2 behavior (after accounting for Session 1 behavior) than replay in the Wake group. This analysis assesses cross-session replay-behavioral relationships, as described above, separately for the Sleep and Wake groups. We found evidence in support of this idea: Session 1 replay predicts Session 2 behavior more strongly in the Sleep group than the Wake group ($t[22]=1.847$, $p=0.039$, one-tailed).”

3. With regard to the experimental task, some details should be clarified. On page 5, it is mentioned that “participants learned about 15 novel “satellite” objects” – what does “about” mean here? Was it not exactly 15 for all subjects, i.e. five of each class?

We appreciate the Reviewer catching this ambiguity in our use of the word “about.” There are always 15 objects; we were using “about” in the sense of “participants learned about the satellites.” We have now updated the sentence to avoid this ambiguity:

“Participants learned the features of 15 novel “satellite” objects organized into three classes (Fig. 1).”

Also, did the learning satellites include the prototype?

Yes. We now make this clear:

“One of the learned satellites in each category is the prototype (shown on the left for each category in Fig. 1a), which contains all the prototypical parts for that category.”

When subjects were required to “reconstruct the satellite images from scratch” during the training session, was this done for all of the 15 learned satellites?

Yes. We now include this information:

“In the first phase, which lasted 15 minutes on average, satellites were introduced one by one, with each of the 15 satellites shown once.”

4. In the methods section, it says that the training session continued until the learning criterion of 66% correct trials was reached or until 60 minutes had passed. How many subjects did not reach the criterion within the 60 minutes? Do the results differ for these subjects?

We appreciate the Reviewer bringing this up, as we realize leaving out this information left open the possibility that a substantial number of participants had trouble with the training. There was actually just one participant who didn't meet this criterion, and that person was very close to 66%, at an accuracy of 63%. We now include this information:

“This phase of training continued until the participant reached a criterion of 66% of trials correct on a block of 32 trials, or until 60 minutes had passed. Only one participant did not reach the criterion, but was very close, with an accuracy of 63%.”

Reviewer #3 (Remarks to the Author):

Schapiro and colleagues present compelling new fMRI data suggesting how memories might be selected for replay in humans. Participants were trained to identify different “classes” of satellites, which had both shared (common to all satellite exemplars within a class) and unique (specific to one particular satellite exemplar) features. Memory and neural replay of specific satellites were both measured during the first session, and again during session a second session ~12 hours later. Half of the participants slept between sessions (session 1 in PM, session 2 in AM), and the other half did not (session 1 in AM, session 2 in PM). The main findings were that (1) the sleep group showed less forgetting of unique visual features and improved memory for the shared visual features as compared with the wake group, (2) more replay was observed in hippocampus for items remembered less well during session 1 and items remembered better during session 2, and (3) replay of specific items was associated with memory improvement (S2-S1 performance) only in the sleep group. These findings suggest that (among other things) weak memories receive preferential replay during awake rest periods, and that the effects of awake replay may be enhanced when a delay includes sleep.

These are very exciting data from a carefully conducted study. Of note, all of the replay analyses are on an item-by-item basis, such that the authors are able to make claims about which specific memories (within a given participant) show evidence of replay. This feature makes the results especially timely and interesting, and I believe they will be widely influential in the field—memory replay is a hot topic, and this study fills a critical gap in describing one factor (memory strength) that dictates what is deserving of replay, and what is not. The statistical analyses are appropriate and valid. I am generally very enthusiastic about this paper; however, I do have a few questions and concerns:

We appreciate the Reviewer's enthusiasm and their very useful comments and suggestions.

1) I am left wondering the extent to which the order of replay versus test might determine the direction of the relationship between memory and replay. My intuition is that the impact of test on replay (first session) and the impact of replay on test (second session) could be the result of very different neurobiological mechanisms, and that might be part of the reason for the change in direction from session 1 to 2. For example, it could be the case that at the session 1 test, satellites on which the participant is incorrect are tagged for later replay (it seems to me that this could happen despite no external feedback). The authors address this possibility in the discussion. For session 2, it could be the case that satellites that are replayed are primed, and thus easier to access during the subsequent test. The item representation or “template” phase also serves as a reminder that could influence this process, which I assume is by design. Could the authors touch on these possibilities? (Related to this point, I quite liked the inclusion of the attention univariate control analysis, which seems to suggest participants aren't simply paying attention to/studying those less

familiar or “weaker” satellites more during the template phase.)

Reviewer 1 raised very similar points in their first comment, and we regret not being clearer in the previous draft that we *do* think the order of replay versus test is an important determinant of our results. As the Reviewer is describing, we think the most likely explanation of the results is that satellites that the participant has trouble remembering are tagged for later replay, even without external feedback (a point which we now make clearer in the Discussion section “*Replay prioritizes weaker memories*”). Then replay increases the likelihood of subsequent recall. We do not view these interpretations as alternative possibilities, but instead as the best interpretation of the data. See our response to Reviewer 1 for additional comments and text changes.

The template measurement period was actually not intended to serve as a reminder. We had to decide whether the rest period would always occur adjacent to the memory test or whether the measurement period and rest period would always occur in the same order. Each option has pros and cons, and we decided on the latter option. This option introduces what we view to be an undesirable reminder of the satellites between the first memory test and first rest period. However, the reminder is the same across all satellites, so we do not think this biases the results, which rely on differential patterns across satellites. There could be bias introduced by differential attention, but as the Reviewer appreciated, our control analysis is helpful in ruling that out. In introducing that control analysis, we now explain:

“The experiment was designed such that the measurement and rest periods occurred in the same order in both sessions, in case the order of these two experiment elements might have consequences for the detection of replay events. This design decision required the measurement period to come between the memory test and rest period in Session 1, which introduces the possibility that unassessed new learning could occur during the measurements. This leads to a potential alternative explanation for the negative correlation between memory and subsequent replay in Session 1...”

2) I would be interested to hear the authors’ thoughts on where these “weak” memories might lie on the continuum of strength in terms of everyday memory. The authors find that the weaker of the satellite memories showed preferential replay, but it strikes me as a key point that this “weak” is relative to the other satellites—all of which were more well-learned than many real world, one-off experiences. I presume there is a lower limit on this selection process—i.e., memories that are too weak might not be replayed at all (or may be tagged but too weak to be reactivated), whereas medium strength memories could be the prime target. It might be helpful, especially for a casual reader who does not dig into the methods of the present paper, to describe a bit about what is meant by “weak” in the grand scheme of things.

This is an excellent point – the paper would definitely benefit from comments along these lines. We have now added the following to the Discussion section:

“It seems unlikely that the relationship between memory strength and replay likelihood is strictly monotonic. If an experience is barely encoded at all, there may not be a strong enough representation of that experience to support any replay. With respect to the full range of encoding strength possible in everyday life, ours and other studies with many learning trials are likely sampling from the medium to high range of memory strength. It would be interesting for future work to map out a wider range of memory strength to assess whether replay likelihood is indeed an inverted U-shaped function of strength.”

3) I believe that through the end of session 1, the experience of participants in the two groups was

identical (with the exception of time of day). If I am correct, I believe any (numerical, even if not reliable) differences during session 1 (e.g., Figure 4B) are not meaningful. I apologize if this was in the paper and I missed it, but it would be useful to emphasize in the paper that for analyses restricted to session 1, no differences between groups would be expected.

Correct, the only difference between the two groups in Session 1 was time of day. For the behavioral results, we note:

“Performance on the first test was not different for subjects in the Sleep vs. Wake groups in unique, shared, or novel item features ($p>0.564$), suggesting that time of day does not influence performance on this task (as in our previous work with this paradigm [30]).”

For the only neural analysis that separates groups and looks only at Session 1 data, we updated a comment about time of day to be clearer in this regard:

“In both sessions, left was not different than right hippocampus (Session 1: $t[23]=0.773$, $p=0.447$; Session 2: $t[23]=1.453$, $p=0.160$), the Sleep group was not different than Wake (Session 1: $t[22]=0.368$, $p=0.717$; Session 2: $t[22]=0.764$, $p=0.453$), verbal was not different than visual (Session 1: $t[22]=1.001$, $p=0.328$; Session 2: $t[21]=0.756$, $p=0.458$), and shared was not different than unique (Session 1: $t[23]=0.203$, $p=0.841$; Session 2: $t[20]=0.693$, $p=0.496$). Note that the lack of difference between Sleep and Wake groups in Session 1 suggests that time of day did not influence replay.”

4) Eyeballing the charts, it looks like the effects might be right-lateralized in the wake group, and more bilateral in the sleep group. Was this the case, and if so, is that meaningful?

In response to Reviewer 2’s first comment, we ran mixed effects models to assess interactions between factors. We did not find any reliable interactions between group and hemisphere. However, this is not necessarily the most direct test of the Reviewer’s question about laterality. To test this more directly, we computed the difference between each subject’s left and right hippocampal correlations, which provides an index of the degree of laterality. These indices were marginally different in the Sleep and Wake groups (Session 1: $t[22]=1.824$, $p=0.082$; Session 2: $t[22]=1.929$, $p=0.067$). Since the results are not statistically significant and the interaction did not come up in the mixed effects model, we have decided not to include this in the paper.

5) Given the resolution of the functional data (3x3x3mm), subfield analyses are not interpretable. In my opinion, these should not be performed at all at this resolution; if the authors wish to keep them in, I would suggest adding extensive caveats throughout the paper.

The Reviewer is right that these results were not presented with sufficient caution, given the data resolution. While not central to the paper, we do think it is worthwhile to report these results, as they are consistent with our recent model of the function of hippocampal subfields and may be useful in the future as findings are aggregated across studies. It is true that by themselves, these data do not provide strong evidence regarding subfield function. The results are restricted to one paragraph of the Results section – we did not bring them up in the discussion. We have now added a caveat to that paragraph of the Results so that readers will understand the limitations:

“These subfield results should be treated as exploratory, as our functional scanning resolution was too low to permit confident assessment of subfield activity.”

6) On p. 16, the authors write: “One study in rodents found that replay during rest was important

for gradually-strengthened but not quickly-formed memories [6] ... , perhaps reflecting similar processes.” It is not clear to me from this wording why “gradually strengthened” memories would be weaker and replayed more. Could the authors expand on this point or reword?

This is a great point. It does not follow that gradually learned information is necessarily weaker. We appreciate the Reviewer catching this out and we have decided to simply remove this point.

7) Clarification point: Were the same features tested for a given satellite for session 1 and session 2 (i.e., were the test trials identical with the exception of the trial order)?

Yes, we appreciate the Reviewer pointing out this ambiguity. We have now clarified the wording:

“Then they got out of the scanner and completed the same test phase as in the first session, with identical trials presented in a different random order.”

Reviewer #1:

The authors did a commendable job revising the manuscripts, and I believe the revisions have greatly strengthened the paper. I was especially excited to see the results of the item-specific, within-category results. However, I still have a few lingering concerns about the manuscript which I detail below.

Comment 1: In the revised manuscript, the authors greatly clarified the goals of their study design. I can see how what I considered a confound, was actually a purposeful design element. However, I think language needs to be added throughout the manuscript to discuss the results as a function of this specific behavioral context. For example in the abstract, there is no mention of the relationship of rest periods to the test, and thus the findings seem like they are purely about the delay rather than the proximal relationship to test. Similarly, in the final paragraph of the introduction, the authors state what the design allowed them to study without mentioning the position of the rest in relationship to test. I think this layer of specificity throughout the manuscript would better frame the results, and help the findings fit into a larger literature on reactivation. Similarly, I believe the title should be modified to better reflect this study goal.

We agree that there is room for further clarity about the relationship between rest periods and behavioral tests. The abstract now reads:

“We used fMRI pattern analysis to track item-level replay in the hippocampus during an awake rest period after participants studied 15 objects and completed a memory test. Objects that were remembered less well were replayed more during the subsequent rest period, suggesting a prioritization process in which weaker memories—memories most vulnerable to forgetting—are selected for replay. In a second session 12 hours later, more replay of an object during a rest period predicted better subsequent memory for that object.”

The end of the introduction now spells out the rest-behavior timing logic. We believe making this explicit at the end of the introduction will be effective in preventing confusion while reading the rest of the paper:

“In Session 1, we (1) taught participants the features of the satellites, (2) tested their memory for these features, (3) measured the neural response generated by each of the satellites in the fMRI scanner, and then (4) used pattern analysis to assess whether individual items were replayed by the hippocampus during a rest period in the scanner. In the second session, we again measured neural responses to the individual satellites and assessed replay during rest. After the rest period, we tested memory for the satellites a second time. Note that in Session 1, the memory test preceded the rest period whereas in Session 2, the rest period preceded the memory test.

This design allowed us to answer four questions critical to understanding the role of hippocampus in the consolidation of object knowledge:

1. Are representations of recently learned individual items replayed in the human hippocampus during quiet rest? Prior literature in humans and rodents suggests that this occurs, but it has not yet been observed at the resolution of individual items in humans.

2. In Session 1, where the memory test precedes the rest period, does probability of replay during rest relate to the strength of a memory? It is possible that weaker memories are

prioritized for replay, as suggested by the sleep literature; alternatively, stronger memories may be more likely to persist into subsequent rest.

3. In Session 2, where the rest period precedes the memory test, does replay of individual items in the hippocampus predict subsequent memory? Prior literature in humans and rodents suggests a positive relationship, though this has not yet been assessed for individual hippocampal memories in humans.

4. Does the relationship between wake replay and memory improvement across the delay between sessions relate to the presence of intervening sleep? Replay measured during awake rest periods may reflect (or perhaps influence^{33,34}) the processing that continues to occur in the intervening period between sessions, and this processing may be especially beneficial over sleep³⁵.”

We also improved the description of the protocol in the Introduction:

“In Session 1, we (1) taught participants the features of the satellites, (2) tested their memory for these features, (3) measured the neural response generated by each of the satellites in the fMRI scanner, and then (4) used pattern analysis to assess whether individual items were replayed by the hippocampus during a rest period in the scanner. In the second session, we again tested memory, measured neural responses to the individual satellites, and assessed replay during rest, with the one difference that the memory test came last (after the rest period) instead of first.”

In case the logic from the introduction was missed, the first paragraph of the Discussion is now very clear in this regard:

“To address these questions, we taught participants the features of 15 objects and assessed replay of individual object representations before and after memory tests. We did extensive repeated exposure of the objects during fMRI scanning to maximize our ability to create high-fidelity templates that could be used to track replay. We found in the first session, where a memory test preceded a rest period, that objects that were initially remembered poorly were subsequently replayed more in the hippocampus, while in the second session, where the rest period preceded the memory test, that objects that were replayed more were subsequently better remembered.”

We believe that the title is an accurate and clear description of the most important results of the study, so we prefer not to change it.

Comment 2: I think my prior concern about the different memory conditions was that the design of the study seemed to reflect a goal to make comparisons across exemplars and category-level structure. However, there is very little discussion of this aspect of the task or how the findings relate to this differentiation. In brief, it would be helpful to have a section in the discussion detailing how this study is consistent or inconsistent with the findings and theoretical framework discussed in their 2017 paper.

We appreciate the prompt to include more discussion about how these findings relate to those from our 2017 paper. We believe that our current results are nicely consistent with the previous results, and it is useful to highlight this. The relevant part of the Discussion section now reads:

“Consistent with a prior study³⁰, memory for the satellite objects, and in particular the features shared across members of a category, was better after a night of sleep. These results may reflect an increased reliance on cortical areas as a result of systems consolidation⁴⁸, as cortical areas employ more overlapping representations than the hippocampus and should therefore facilitate memory for shared features^{30,49}. One cortical region often implicated as a consolidation site is the mPFC⁵⁰, and our whole-brain univariate analysis found increased activation across sessions in that region for the Sleep but not the Wake group. We did not find evidence for multivariate changes—in replay or representational similarity—in cortical regions, which may mean multivariate changes did not occur on this timescale, or it may simply be a null result due to methodological or power limitations. The univariate finding suggests that the mPFC is overall more engaged by the satellites after a period of sleep, but leaves open whether there are changes in its representational content. We did not find any evidence that the hippocampus was less involved in this task from the first to second session in the Sleep group c.f. ⁵¹, though any decreased reliance on the hippocampus may take more time.

We did not find that hippocampal replay had any differential relationship with memory for shared vs. unique features of the satellites (nor for verbal vs. visual features). This result is consistent with the observation in this study and in our previous study³⁰ that there is no interaction in memory change across sessions between Sleep vs. Wake group and unique vs. shared features. This suggests that there is coherent replay of the entire object representation that does not favor particular properties. While both feature types appear to benefit equally from replay and from sleep, it may be, as mentioned above, that systems consolidation during sleep allows shared features to undergo and maintain a memory boost above the pre-sleep baseline. The lack of specificity in the relationship between replay and memory for particular feature types also suggests that these results are likely to generalize to other forms of item memory. Broadly, we expect our findings to generalize to any two-test paradigm where replay is measured shortly after the initial test and shortly before the delayed test.”